# On the Difficulty of Learning in Classification Problems: Optimality and Information-Theoretic Perspectives

## Abstract

This paper studies the hardness of learning in classification tasks. We formulate a classification problem using a fixed input distribution and a variable ground-truth classifier drawn from a prior distribution, and consider an average notion of risk measure. We then derive a closed-form solution for the optimal learner and the optimal risk, and use the latter to measure the hardness of learning. Using Fano's Inequality, we establish a risk lower bound in terms of information-theoretic quantities. Our bound overcomes the over-pessimism of classical lower bounds in statistical learning theory. Comparing with existing information-theoretic lower-bounds in similar settings, our bound is tighter and more practically relevant. Our analysis reveals a tradeoff between two key quantities that govern the difficulty of learning in classification problems, which we refer to as *identifiability* and *agreement*. We also characterize the convergence behavior of our lower bound with respect to the sample size.

## 1 Introduction

The rise of deep learning has reshaped the landscape of machine learning and AI, and this methodology has demonstrated enormous successes in numerous areas of machine learning. As a significant research effort has been spent on understanding the power of deep learning (Zhang et al., 2016; Jacot et al., 2018; Belkin et al., 2020; Wang & Mao, 2024), we are interested in an "opposite" research direction, namely, investigating whether there are *practical* learning problems that are hard enough for *all learning algorithms*, including deep learning algorithms. This boils down to studying the hardness of learning problems, and in this paper we primarily focus on classification problems.

In the classical PAC-learning framework (Valiant, 1984), the hardness of learning is studied via lower bounds of achievable risks (Ehrenfeucht et al., 1989; Hanneke, 2016; Goar & Yadav, 2024). In this approach, a carefully designed bad input distribution is used to develop a risk lower bound so that every classifier (or "concept") in the considered hypothesis class gives rise to a risk value larger than the bound. Another type of lower bound is via minimax risks (Antos & Lugosi, 1996; Boucheron et al., 2005; Jiao et al., 2015; Malach & Shalev-Shwartz, 2022; Ma et al., 2024), in which the risk of the best classifier (or of the best learning algorithm) for the worst data distribution is used as the lower bound. A key limitation of these bounds is that the employed distributions are often far from those of practical interest, and hence the resulting lower bounds, as a proxy of hardness, are too pessimistic to be practically relevant.

Another class of lower bounds rely on tools from information theory (Zhao et al., 2013; Chen et al., 2016; Scarlett & Cevher, 2019; Jeon & Roy, 2022; Morishita et al., 2022; Dong et al., 2025), particularly Fano's Inequality (Verdú et al., 1994; Yu, 1997; Cover & Thomas, 2006). In this line of works, instead of treating each candidate ground-truth classifier in the concept class with equal footing, one assumes a prior distribution on the space of hypotheses, which indicates the probability (mass or density) of a concept being selected. The risk is defined as an average with respect to this distribution and a risk lower bound in terms of some information-theoretic quantities is obtained via Fano's Inequality.

To address the limitation of the PAC-learning framework, this work considers a formulation in line with the second class of works, while focusing on classification settings. Specifically, a classification

problem is specified by a pair $(\mu, \mathcal{E}_F)$, where $\mu$ is the input distribution, modeling a distribution arising in practice, and $\mathcal{E}_F$ is a distribution of the hypotheses, modeling the learner's prior knowledge of the ground-truth. We define the overall risk of a learner as its average classification error (see Section 3 for a precise formulation).

In this setting, we derive closed-form solutions of the optimal learner and the optimal risk, and use the latter to measure the hardness of learning. By carefully inspecting the relationship between the involved random variables, we adopt a different application of Fano's Inequality to derive a risk lower bound. We also theoretically investigate the asymptotic convergence behavior of our risk lower bound with respect to the size of training sample.

Particularly relevant to the topic of this paper, i.e. classification problems, are the work of Jeon & Roy (2022) and that of Chen et al. (2016), where the problems setups share great similarity with the present paper. However, Jeon & Roy (2022) uses KL divergence as a measure of risk , making the resulting lower bound largely irrelevant for practical considerations, which will be discussed in Remark 1. On the other hand, the lower bounds in Chen et al. (2016), albeit derived for very general settings, are in fact quite weak for classification problems considered in this paper —we now demonstrate the weakness of the bounds in Chen et al. (2016) using the following toy example; more discussion can be found in Section 4.4.

**Example 1.** Consider a binary classification problem with label space $\mathcal{Y} = \{-1, +1\}$. Let the input $X$ be drawn from the uniform distribution on $[0, 1]$. Suppose that the ground-truth classifier is chosen equally likely from $\{f_a, f_b\}$, where $f_a$ and $f_b$ are defined as follows,

$$f_a(x) = \begin{cases} -1 & x \in [0, \frac{1}{2} - \frac{1}{2}\epsilon) \\ +1 & x \in [\frac{1}{2} - \frac{1}{2}\epsilon, 1] \end{cases} \quad \text{and} \quad f_b(x) = \begin{cases} -1 & x \in [0, \frac{1}{2} + \frac{1}{2}\epsilon) \\ +1 & x \in [\frac{1}{2} + \frac{1}{2}\epsilon, 1] \end{cases} \quad (1)$$

for some $\epsilon \in (0, 1)$, as shown in Figure 1. We are interested in the hardness of learning in this setting, measured by the the best achievable classification error. In fact, the optimal risk can be derived in closed form as (the detailed computation is provided in Appendix A.1):

$$R^* = \epsilon (1 - \epsilon)^n / 2, \quad (2)$$

where $n$ denotes the size of the training sample. Theorem 1 of this paper (see Section 4.3) lower-bounds the optimal risk by $\epsilon^2 (1 - \epsilon)^{2n} / 4$; the lower bound given by Chen et al. (2016) is however 0, completely vacuous.

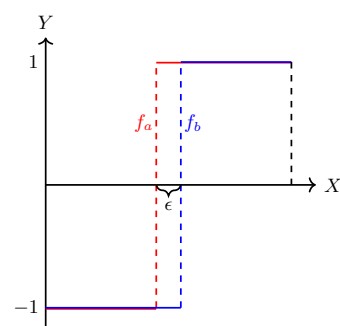

Figure 1: Toy Example 1.

The comparison between the bound of Chen et al. (2016) and that of ours in this example also reveals an interesting interplay between "identifiability" and "agreement" that governs the hardness of a learning problem. Notice that in this example, as long as one point in the interval $[\frac{1-\epsilon}{2}, \frac{1+\epsilon}{2})$ is observed, $f_a$ and $f_b$ can be distinguished. The larger $\epsilon$ is, the easier it is to identify the ground-truth classifier. That is, if $\epsilon$ is large, the ground-truth will be identifiable and hence the error will be small. If $\epsilon$ is small, it becomes unidentifiable. However, *identifiability* alone does not determine the achievable classification error. Specifically, consider a small $\epsilon$. In this case, $f_a$ and $f_b$ highly agree in their predictions. Then even when we decide on a wrong classifier, we will still achieve a low population classification error, making the learning problem not difficult. In other words, the *agreement* between the members in the hypothesis class also plays a role in the difficulty of learning.

The lower bound of Chen et al. (2016), when restricted to a classification setting is effectively governed by the conditional entropy $H(F \mid S^n)$, where $F$ denotes the variable ground truth classifier and $S^n$ denotes the labeled training sample of size $n$. Note that this conditional entropy merely reflects the identifiability among hypotheses but fails to capture their agreement. Unlike Chen et al. (2016), our bound is governed by the quantity $H(F \mid S^n) - H(F \mid S^{n+1})$, which represents the additional knowledge of $F$ obtained from one extra training example beyond $S^n$. Notice that this quantity is upper bounded by $H(F \mid S^n)$, indicating that it still captures identifiability. On the other hand, its scale is determined by the degree of agreement among hypotheses on a single example. Thus,

by incorporating both identifiability and agreement, our bound provide a stronger approximation of the difficulty of learning than Chen et al. (2016). Our analysis thus reveals that an interesting tradeoff may exist between the two quantities.

**Notations.** Throughout the paper, we use calligraphic capitalized letters to denote sets, capital letters to denote random variables, and lowercase letters to denote realizations of the corresponding random variables. For any measurable set $\mathcal{A}$, we denote by $\Delta(\mathcal{A})$ the set of all probability distributions on $\mathcal{A}$. We use $\Omega(p)$ to denote the support of distribution $p$. For any $a \in \mathcal{A}$, $\delta_a \in \Delta(\mathcal{A})$ is the distribution that places the entire probability mass 1 on $a$. We will often use the symbol $\mathbb{P}$ to denote a probability distribution, with its subscript indicating the involved random variables.

The remainder of this paper is organized as follows. In Section 3, we present the classification problem formulation, describing how a sample is generated and how a learner operates after observing the sample. We also introduce necessary notations and define the risk functions in this section. Section 4 presents two types of lower bounds for a given classification problem, and we interpret the bounds from a practical perspective by analyzing a toy example. In Section 5, we conduct a convergence analysis with respect to the sample size, characterizing the rate at which the lower bound converges as the sample size goes to infinity. Finally, Section 6 concludes this paper and discusses the limitations of this work. The proofs of our main theoretical results and further discussions of related works can be found in the Appendix.

## 2 RELATED WORKS

In this section, we will formally state the main results from Chen et al. (2016) and Jeon & Roy (2022), whose work share great similarities with ours.

In Jeon & Roy (2022), the authors have formulated any supervised learning problem as follows. Let $\mathcal{X}$ and $\mathcal{Y}$ be an input space and a label space respectively, note that $\mathcal{Y}$ needs not to be finite. Denote by $\mu$ a fixed distribution on $\mathcal{X}$. Let $\mathcal{F} := \{f : \mathcal{X} \to \Delta(\mathcal{Y})\}$ be a set of labeling functions, each of which returns a distribution on $\mathcal{Y}$ given input. Let $\mathcal{E}$ be a distribution on $\mathcal{F}$. For any $F \sim \mathcal{E}$, a learner observes a training sample $S^n := \{(X_1, Y_1), \cdots, (X_n, Y_n)\}$. Each pair $(X_i, Y_i)$ is i.i.d. sampled via $X_i \sim \mu$ and $Y_i \sim F(X_i)$. The learner uses an algorithm $\mathscr{A} : (\mathcal{X} \times \mathcal{Y})^n \to \Delta(\mathcal{F})$ to learn a distribution $\mathscr{A}(S^n)$ on $\mathcal{F}$. The measure of risk is defined in terms of KL-Divergence as follows.

$$R_{\mathrm{KL}}(\mathscr{A}; \mu, \mathcal{E}) := \underset{f \sim \mathcal{E}}{\mathbb{E}} \underset{s^n}{\mathbb{E}} \underset{\hat{f} \sim \mathscr{A}(s^n)}{\mathbb{E}} \underset{x \sim \mu}{\mathbb{E}} \left[ d_{\mathrm{KL}} \left( f(x) \| \hat{f}(x) \right) \right] \tag{3}$$

The main theorem of Jeon & Roy (2022) states that the optimal risk $R_{\mathrm{KL}}^* := \min_{\mathscr{A}} R_{\mathrm{KL}}$ can be exactly characterized in terms of conditional mutual information as follows,

$$R^* = I\left(F; (X, Y) \mid S^n\right). \tag{4}$$

On the other hand, Chen et al. (2016) has formulated learning problems from a point of view of parameter estimation. In their framework, $\mathcal{F}$ can be regarded as a parameter space. For any $F \sim \mathcal{E}$, a training sample $S^n$ is sampled from a distribution $\mathbb{P}_F$ that is uniquely determined by $F$. Upon observing $S^n$, an algorithm $\mathscr{A} : (\mathcal{X} \times \mathcal{Y})^n \to \mathcal{F}$ will return an estimation of $F$. Let $L : \mathcal{F} \times \mathcal{F} \to \mathbb{R}_{\geq 0}$ be a non-negative loss function, then the risk is defined as follows,

$$R_L(\mathscr{A}; \mathcal{E}) := \underset{f \sim \mathcal{E}}{\mathbb{E}} \underset{s^n}{\mathbb{E}} L\left(f, \mathscr{A}(s^n)\right). \tag{5}$$

For any $p \in [0, 1]$, denote by $\phi(p) := \frac{1}{2} \log \frac{1}{2p} + \frac{1}{2} \log \frac{1}{2(1-p)}$ the binary KL-Divergence from $[1/2, 1/2]^\top$ to $[p, 1-p]^\top$. Chen et al. (2016) has proved that the optimal risk $R_L^* := \min_{\mathscr{A}} R_L$ can be lower bounded as follows,

$$R_L^* \geq \frac{1}{2} \sup \left\{ t > 0 : \sup_{\hat{f} \in \mathcal{F}} \mathcal{E}\left( f \in B_t\left(\hat{f}, L\right)\right) < 1 - \phi^{-1}\left(I\left(F; S^n\right)\right) \right\}, \tag{6}$$

where $B_t\left(\hat{f}, L\right) := \left\{ f \in \mathcal{F} : L\left(f, \hat{f}\right) < t \right\}$ and $\phi^{-1}$ is the inverse of $\phi$. They have also proposed the following variational bound, which is more computable.

$$R_L^* \geq \frac{1}{2} \sup \left\{ t > 0 : \sup_{\hat{f} \in \mathcal{F}} \mathcal{E}\left( f \in B_t\left(\hat{f}, L\right)\right) < \frac{1}{4} \exp\left(-2I\left(F; S^n\right)\right) \right\}. \tag{7}$$

In particular, if $\mathcal{Y}$ is a finite set and the loss function $L$ is defined as

$$L\left(f, \hat{f}\right) := \mathop{\mathbb{E}}_{x \sim \mu} \mathop{\mathbb{E}}_{y \sim f(x)} \mathop{\mathbb{E}}_{\hat{y} \sim \hat{f}(x)} \mathbb{1}\{y \neq \hat{y}\}, \tag{8}$$

and the formulation of Chen et al. (2016) reduces to ours—presented in Section 3—as a special case.

## 3 PROBLEM FORMULATION

We consider classification problems with an input space $\mathcal{X}$ and a finite output space (or label space) $\mathcal{Y}$. In this context, a *soft classifier* is a function $f$ mapping $\mathcal{X}$ to $\Delta(\mathcal{Y})$, where for every label $y \in \mathcal{Y}$, the $y^{\text{th}}$ component $f_y(x)$ of $f(x)$ is the probability that $f$ assigns label $y$ to input $x$. We may also write $f(y|x)$ in place of $f_y(x)$. The space of all soft classifiers is denoted by $\mathcal{F}$. When the output $f(x)$ of a soft classifier $f$ is a one-hot distribution for every input $x$, i.e., $f(x)$ is $\delta_{y(x)}$ for some label $y(x)$ that depends on $x$, we say that $f$ is a *hard classifier*[1]. Notably, every soft classifier $f$ can be "hardened" into a hard classifier $f^{\text{H}}$, where $f^{\text{H}}(x) := \delta_{\arg\max_{y \in \mathcal{Y}} f(y|x)}$ for every $x$. We then formulate classification problems as follows.

**Model of Nature.** Let $\mu \in \Delta(\mathcal{X})$ be a distribution on $\mathcal{X}$ and $\mathcal{E}_F \in \Delta(\mathcal{F})$ be a distribution on $\mathcal{F}$. Nature first draws a ground-truth classifier $f$ from $\mathcal{E}_F$. It then samples $s_{\mathcal{X}}^n := \{x_i\}_{i=1}^n$ i.i.d. from $\mu$ and assigns a label $y_i$ by sampling from $f(x_i)$ for each $i = 1, \cdots, n$. We denote $\{y_i\}_{i=1}^n$ by $s_{\mathcal{Y}}^n$. For notational simplicity, we may denote the sampling process of $s_{\mathcal{Y}}^n$ from $s_{\mathcal{X}}^n$ and $f$ by $s_{\mathcal{Y}}^n \sim f(s_{\mathcal{X}}^n)$. Nature then reveals the training sample $(s_{\mathcal{X}}^n, s_{\mathcal{Y}}^n)$ to the learner.

**Model of Learner.** A learner $\mathscr{A}$ is a function mapping $(\mathcal{X} \times \mathcal{Y})^n$, for any sample size $n$, to $\Delta(\mathcal{F})$. Upon observing training sample $(s_{\mathcal{X}}^n, s_{\mathcal{Y}}^n)$, the learner outputs a distribution $\mathscr{A}(s_{\mathcal{X}}^n, s_{\mathcal{Y}}^n)$ over $\mathcal{F}$. With a slight abuse of notation, for any $\hat{f} \in \mathcal{F}$, we use $\mathscr{A}(\hat{f} \mid s_{\mathcal{X}}^n, s_{\mathcal{Y}}^n)$ to denote the probability density (or mass) assigned to $\hat{f}$ under $\mathscr{A}(s_{\mathcal{X}}^n, s_{\mathcal{Y}}^n)$, analogous to the notation $f(y \mid x)$ for a soft classifier. For each new input $x$ drawn from $\mu$, the learner draws $\hat{f}$ from $\mathscr{A}(s_{\mathcal{X}}^n, s_{\mathcal{Y}}^n)$ and samples from $\hat{f}(x)$ a predicted label for $x$.

**Performance Metrics.** Given a learner $\mathscr{A}$, we define its *risk* with respect to any input distribution $\mu \in \Delta(\mathcal{X})$ and any classifier $f \in \mathcal{F}$ by

$$R(\mathscr{A}; \mu, f) := \mathop{\mathbb{E}}_{\substack{s_{\mathcal{X}}^n \sim \mu^n \\ s_{\mathcal{Y}}^n \sim f(s_{\mathcal{X}}^n)}} \mathop{\mathbb{E}}_{\hat{f} \sim \mathscr{A}(s_{\mathcal{X}}^n, s_{\mathcal{Y}}^n)} \mathop{\mathbb{E}}_{x \sim \mu} \mathop{\mathbb{E}}_{\substack{y \sim f(x) \\ \hat{y} \sim \hat{f}(x)}} \mathbb{1}\{y \neq \hat{y}\}. \tag{9}$$

The *overall risk* of the learner $\mathscr{A}$ for a pair $(\mu, \mathcal{E}_F)$ is then defined as

$$R(\mathscr{A}; \mu, \mathcal{E}_F) := \mathbb{E}_{f \sim \mathcal{E}_F} \left[ R(\mathscr{A}; \mu, f) \right] \tag{10}$$

In this formulation, a classification problem is completely specified by a pair $(\mu, \mathcal{E}_F)$. The objective of the classification problem is then to find a learner $\mathscr{A}$ that minimizes the overall risk.

With this, we have completed the formulation of the classification problem. Figure 2 presents the relationships among the involved random variables, illustrated as a Bayesian network (Pearl, 1988).

This formulation differs from that in PAC-learning (Valiant, 1984) in several ways. First, the uncertainty of the ground-truth classifier is modeled as a distribution over $\mathcal{F}$, rather than a *subset* of $\mathcal{F}$ as in PAC-learning, where each classifier is treated with equal footing. Second, we use an average notion of risk rather than the worst-case risk as in PAC-learning. These two differences allow this formulation to de-emphasize the pessimistic effect of those

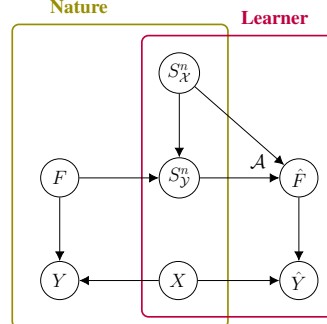

Figure 2: The relationship of random variables in our formulation. $F$: ground-truth classifier. $\hat{F}$: learned predictor.

---

[1]Practitioners usually treats a hard classifier as a function mapping $\mathcal{X}$ to $\mathcal{Y}$ so that $f(x)$ is some $y(x)$. Our treatment is equivalent and offers some notational convenience.

"bad" but rare ground-truth classifiers. Finally, this formulation restricts to a fixed input distribution (corresponds to those arising in reality), rather than an unconstrained family in PAC-learning, which includes very bad but unrealistic input distributions. This allows the resulting lower bounds in this framework to be less pessimistic, better reflecting the difficulty of learning arising in practice.

**Remark 1.** *Note that the theoretical formulation in Jeon & Roy (2022) differs from ours in the choice of performance metric. Specifically, in Jeon & Roy (2022), the risk (when restricted to classification settings) of a learned classifier $\hat{f}$ with respect to the ground-truth classifier $f$ on an input $x$ is measured via the KL divergence between $f(x)$ and $\hat{f}(x)$. When $\hat{f}$ and $f$ are hard classifiers and when an error occurs, the KL divergence explodes to infinity. This makes their formulation inappropriate for classification settings.*

## 4 OPTIMAL LEARNER, OPTIMAL RISK, AND LOWER BOUNDS

Given a classification problem $(\mu, \mathcal{E}_F)$, the optimal learner $\mathscr{A}^*$ is one that has the minimal overall risk. That is,

$$\mathscr{A}^* := \underset{\mathscr{A}:(\mathcal{X}\times\mathcal{Y})^n \to \Delta(\mathcal{F})}{\arg\min} R(\mathscr{A}; \mu, \mathcal{E}_F). \tag{11}$$

The optimal risk (also known as the Bayes risk) of the learning problem, which we denote by $R^*(\mu, \mathcal{E}_F)$ or simply $R^*$ when $(\mu, \mathcal{E}_F)$ is clear from context, is the overall risk of the optimal learner, namely, $R^* := R(\mathscr{A}^*; \mu, \mathcal{E}_F)$. Throughout this paper, we will use $R^*$ as the measure of difficulty of learning. In this section, we analyze this quantity and its lower-bound proxy.

### 4.1 POSTERIOR DISTRIBUTION OF CLASSIFIER

Since the support of the input distribution $\mu$ may only cover a subset of the input space $\mathcal{X}$, there may exist multiple classifiers in $\mathcal{F}$ that behave identically on $\Omega(\mu)$ and hence are not distinguishable with respect to $\mu$. The classifiers in $\mathcal{F}$ that behave identically with respect to $\mu$ are then equivalent in this sense. The distribution $\mathcal{E}_F$ on $\mathcal{F}$ thus induces a distribution over the space of such equivalent classes. More precisely, given $\mu$ and for any $f \in \mathcal{F}$, let $\rho_\mu(f)$ denote the restriction of $f$ on the support $\Omega(\mu)$ of $\mu$. Let $\mathcal{F}^\mu$ be the image of $\mathcal{F}$ under the mapping $\rho_\mu$. It is clear that $\mathcal{F}^\mu$ is the space of all functions mapping $\Omega(\mu)$ to $\Delta(\mathcal{F})$. Each member $g \in \mathcal{F}^\mu$ can then be identified with an equivalent class[2] in $\mathcal{F}$. The distribution $\mathcal{E}_F$ on $\mathcal{F}$ then induces a distribution $\mathcal{E}_F^\mu$ on $\mathcal{F}^\mu$ as follows: For every $g \in \mathcal{F}^\mu$,

$$\mathcal{E}_F^\mu(g) = \int_{\mathcal{F}} \mathcal{E}_F(f)\mathbb{1}\{\rho_\mu(f) = g\}\,df. \tag{12}$$

Hereafter, our analysis will be carried out entirely through $\mu$ and $\mathcal{E}_F^\mu$: we only consider classifiers in $\mathcal{F}^\mu$; every occurrence of $f$ refers to a member in $\mathcal{F}^\mu$, not a member in $\mathcal{F}$, although we may still call it a classifier (rather than an equivalent class of classifiers); all probability measures are induced by $\mu$ and $\mathcal{E}_F^\mu$.

For a given classification problem $(\mu, \mathcal{E}_F)$, we have $\mathbb{P}_X(\cdot) = \mu(\cdot)$ and $\mathbb{P}_F(\cdot) = \mathcal{E}_F^\mu(\cdot)$. First, the joint distribution $\mathbb{P}_{F,S_\mathcal{X}^n,S_\mathcal{Y}^n}$ over the ground-truth classifier and the labeled sample is given by:

$$\mathbb{P}_{F,S_\mathcal{X}^n,S_\mathcal{Y}^n}\left(f, s_\mathcal{X}^n, s_\mathcal{Y}^n\right) = \mathbb{P}_F(f)\mathbb{P}_{S_\mathcal{X}^n}\left(s_\mathcal{X}^n\right)\mathbb{P}_{S_\mathcal{Y}^n|S_\mathcal{X}^n,F}\left(s_\mathcal{Y}^n \mid s_\mathcal{X}^n, f\right) = \mathcal{E}_F^\mu(f)\prod_{i=1}^{n}\mu(x_i)f\left(y_i \mid x_i\right). \tag{13}$$

Under the joint distribution $\mathbb{P}_{F,S_\mathcal{X}^n,S_\mathcal{Y}^n}$, the marginal distribution $\mathbb{P}_{S_\mathcal{X}^n,S_\mathcal{Y}^n}$ over the sample is achieved directly by integrating out the classifier $F$:

$$\mathbb{P}_{S_\mathcal{X}^n,S_\mathcal{Y}^n}\left(s_\mathcal{X}^n, s_\mathcal{Y}^n\right) = \int_{\mathcal{F}^\mu} \mathbb{P}_{F,S_\mathcal{X}^n,S_\mathcal{Y}^n}\left(f, s_\mathcal{X}^n, s_\mathcal{Y}^n\right)df = \int_{\mathcal{F}^\mu} \mathcal{E}_F^\mu(f)\prod_{i=1}^{n}\mu(x_i)f\left(y_i \mid x_i\right)df. \tag{14}$$

---

[2]The equivalence relation $\equiv$ here is defined as follows: $f \equiv f'$ if and only if $\rho_\mu(f) = \rho_\mu(f')$.

Then the condition distribution $\mathbb{P}_{F|S_{\mathcal{X}}^n,S_{\mathcal{Y}}^n}$ of the ground-truth classifier conditioned on the sample is obtained via the Bayes' rule:

$$
\begin{aligned}
\mathbb{P}_{F|S_{\mathcal{X}}^n,S_{\mathcal{Y}}^n}\left(f \mid s_{\mathcal{X}}^n, s_{\mathcal{Y}}^n\right) &= \frac{\mathbb{P}_{F,S_{\mathcal{X}}^n,S_{\mathcal{Y}}^n}\left(f, s_{\mathcal{X}}^n, s_{\mathcal{Y}}^n\right)}{\mathbb{P}_{S_{\mathcal{X}}^n,S_{\mathcal{Y}}^n}\left(s_{\mathcal{X}}^n, s_{\mathcal{Y}}^n\right)} \\
&= \frac{\mathcal{E}_F^\mu(f) \prod_{i=1}^n \mu(x_i) f\left(y_i \mid x_i\right)}{\int_{\mathcal{F}^\mu} \mathcal{E}_F^\mu(f') \prod_{i=1}^n \mu(x_i) f'\left(y_i \mid x_i\right) df'} \\
&= \frac{\mathcal{E}_F^\mu(f) \prod_{i=1}^n f\left(y_i \mid x_i\right)}{\int_{\mathcal{F}^\mu} \mathcal{E}_F^\mu(f') \prod_{i=1}^n f'\left(y_i \mid x_i\right) df'} \\
&:= \mathcal{E}_{F|S_{\mathcal{X}}^n,S_{\mathcal{Y}}^n}^\mu\left(f \mid s_{\mathcal{X}}^n, s_{\mathcal{Y}}^n\right).
\end{aligned}
\tag{15}
$$

We have renamed $\mathbb{P}_{F|S_{\mathcal{X}}^n,S_{\mathcal{Y}}^n}$ as $\mathcal{E}_{F|S_{\mathcal{X}}^n,S_{\mathcal{Y}}^n}^\mu$ here due to its significance and to highlight its dependency on $\mathcal{E}_F^\mu$.

### 4.2 OPTIMAL LEARNERS

Consider a classifier $\bar{f}_{s_{\mathcal{X}}^n,s_{\mathcal{Y}}^n}$ obtained by aggregating the ensemble of classifiers given by posterior distribution $\mathcal{E}_{F|S_{\mathcal{X}}^n,S_{\mathcal{Y}}^n}^\mu\left(\cdot \mid s_{\mathcal{X}}^n, s_{\mathcal{Y}}^n\right)$, namely, for each input $x \in \mathcal{X}$

$$
\bar{f}_{s_{\mathcal{X}}^n,s_{\mathcal{Y}}^n}(x) = \mathbb{E}_{f \sim \mathcal{E}_{F|S_{\mathcal{X}}^n,S_{\mathcal{Y}}^n}^\mu\left(\cdot | s_{\mathcal{X}}^n, s_{\mathcal{Y}}^n\right)} f(x).
\tag{16}
$$

**Lemma 1** (Decomposition of the Overall Risk). *Given a classification problem $(\mu, \mathcal{E}_F)$ and a sample size $n$, the overall risk of any learner $\mathscr{A}$ can be decomposed to*

$$
R(\mathscr{A}; \mu, \mathcal{E}_F) = 1 - \mathbb{E}_{x \sim \mu}\mathbb{E}_{s_{\mathcal{X}}^n,s_{\mathcal{Y}}^n \sim \mathbb{P}_{S_{\mathcal{X}}^n,S_{\mathcal{Y}}^n}} \left\langle \bar{f}_{s_{\mathcal{X}}^n,s_{\mathcal{Y}}^n}(x), \mathbb{E}_{\hat{f} \sim \mathscr{A}(s_{\mathcal{X}}^n,s_{\mathcal{Y}}^n)}\hat{f}(x) \right\rangle,
\tag{17}
$$

*where $\langle \cdot, \cdot \rangle$ denotes the inner product between two vectors.*

The sample distribution $\mathbb{P}_{S_{\mathcal{X}}^n,S_{\mathcal{Y}}^n}$ is given in Eq. 14, and the Bayesian posterior $\mathcal{E}_{F|S_{\mathcal{X}}^n,S_{\mathcal{Y}}^n}^\mu$ is defined in Eq. 15. When $(s_{\mathcal{X}}^n, s_{\mathcal{Y}}^n)$ is clear from context, we may denote $\bar{f}_{s_{\mathcal{X}}^n,s_{\mathcal{Y}}^n}$ by $\bar{f}$ for simplicity. Note that both $\bar{f}(x)$ and $\hat{f}(x)$ are distributions over $\mathcal{Y}$, and can therefore be treated as $|\mathcal{Y}|$-dimensional vectors for which the inner product is well defined.

Lemma 1 allows us to obtain the optimal learner $\mathscr{A}^*$ and its overall risk $R^*$, as we show next.

**Proposition 1** (Optimal Learner). *Given a classification problem $(\mu, \mathcal{E}_F)$, the optimal learner $\mathscr{A}^*$ is given by:*

$$
\mathscr{A}^*(s_{\mathcal{X}}^n, s_{\mathcal{Y}}^n) = \delta_{\bar{f}^{\mathrm{H}}},
\tag{18}
$$

*and the optimal risk is*

$$
R^* = 1 - \mathbb{E}_{x \sim \mu}\mathbb{E}_{s_{\mathcal{X}}^n,s_{\mathcal{Y}}^n \sim \mathbb{P}_{S_{\mathcal{X}}^n,S_{\mathcal{Y}}^n}} \max_{y \in \mathcal{Y}} \bar{f}\left(y \mid x\right).
\tag{19}
$$

Proposition 1 states that the optimal learner is the one that deterministically returns the hardened aggregated classifier $\bar{f}^{\mathrm{H}}$ from the Bayesian posterior $\mathcal{E}_{F|S_{\mathcal{X}}^n,S_{\mathcal{Y}}^n}^\mu\left(\cdot \mid s_{\mathcal{X}}^n, s_{\mathcal{Y}}^n\right)$, and the aggregated classifier $\bar{f}$—which depends solely on $\mathcal{E}_{F|S_{\mathcal{X}}^n,S_{\mathcal{Y}}^n}^\mu$—also governs the optimal risk.

We now treat $R^*$ as the measure of the intrinsic hardness of learning in classification problem $(\mu, \mathcal{E}_F)$ based on a training sample of size $n$. We note that this notion of difficulty is solely of statistical nature and has nothing to do with difficulties arising from computations.

The proofs of Lemma 1 and Proposition 1 are provided in Section C and Section D respectively.

### 4.3 LOWER BOUNDS

Recall the optimal risk in Eq. 19, which is explicitly characterized by the Bayesian posterior $\mathcal{E}_{F|S_{\mathcal{X}}^n,S_{\mathcal{Y}}^n}^\mu$—a quantity that is related to the information gained from the observed data. To interpret such knowledge gain from an information theoretic perspective, we derive a lower bound on the overall risk using Fano's inequality in this section.

According the framework illustrated in Figure 2, it can be verified—e.g., using $d$-separation techniques—that $Y$ and $\hat{Y}$ are conditionally independent given $(S_{\mathcal{X}}^n, S_{\mathcal{Y}}^n, \hat{F}, X)$. That is, these variables form a Markov chain $Y - (S_{\mathcal{X}}^n, S_{\mathcal{Y}}^n, \hat{F}, X) - \hat{Y}$. Based on this structure, we obtain the following form of Fano's inequality.

**Lemma 2.** *Given a classification problem $(\mu, \mathcal{E}_F)$ and sample size $n$, let $K$ denote the size of $\mathcal{Y}$, the following inequality holds:*

$$H\left(Y \mid S_{\mathcal{X}}^n, S_{\mathcal{Y}}^n, \hat{F}, X\right) \leq \mathcal{H}_{\mathrm{b}}\left(R\left(\mathscr{A}; \mu, \mathcal{E}_F\right)\right) + R\left(\mathscr{A}; \mu, \mathcal{E}_F\right) \log(K-1), \tag{20}$$

*where $H$ denotes entropy and $\mathcal{H}_{\mathrm{b}}(p) = -p \log p - (1-p) \log(1-p)$ denotes the binary entropy of the Bernoulli distribution with parameter $p \in [0,1]$.*

By rearranging the terms in Eq. 20 and reformulating the conditional entropy $H\left(Y \mid S_{\mathcal{X}}^n, S_{\mathcal{Y}}^n, \hat{F}, X\right)$, we derive a lower bound on $R\left(\mathscr{A}; \mu, f\right)$ for any learner $\mathscr{A}$ in the following Theorem.

**Theorem 1.** *Given a classification problem $(\mu, \mathcal{E}_F)$ and sample size $n$, then for any leaner $\mathscr{A}$ the overall risk is lower bounded by*

$$\text{if } K > 2 : R\left(\mathscr{A}; \mu, \mathcal{E}_F\right) \geq \frac{\Lambda_{\mu, \mathcal{E}_F} - 1}{\log(K-1)},$$
$$\text{if } K = 2 : R\left(\mathscr{A}; \mu, \mathcal{E}_F\right) \geq \frac{\Lambda_{\mu, \mathcal{E}_F}^2}{4}, \tag{21}$$

*where*

$$\Lambda_{\mu, \mathcal{E}_F} = \underbrace{H(F \mid S_{\mathcal{X}}^n, S_{\mathcal{Y}}^n) - H\left(F \mid S_{\mathcal{X}}^n, S_{\mathcal{Y}}^n, X, Y\right)}_{\text{①}=I\left(F; (X,Y) \mid S_{\mathcal{X}}^n, S_{\mathcal{Y}}^n\right)} + \underbrace{H\left(Y \mid F, X\right)}_{\text{②}}. \tag{22}$$

**Sketch of Proof:** A formal proof of Theorem 1 is given in Section F. Briefly, for $K > 2$, Eq. 21 follows directly from Eq. 20. For $K = 2$, we use the relaxation $\mathcal{H}_{\mathrm{b}}(p) \leq 2\sqrt{p(1-p)}$. The decomposition of $\Lambda_{\mu, \mathcal{E}_F}$ follows immediately from the chain rule of mutual information.

In Eq. 21, the lower bounds are effectively governed by the problem-dependent $\Lambda_{\mu, \mathcal{E}_F}$, which is defined by Eq. 22. To better illustrate how this quantity reflects the problem's hardness, we may interpret the terms as follows.

In Eq. 22, term ② reflects the noise of the ground-truth classifiers, that is, the label $Y$ may still contain uncertainty even when $F$ and $X$ are given, due to the assumption that classifiers in our framework are soft. In the special case where $F$ is a hard one, this uncertainty vanishes and $H\left(Y \mid F, X\right) = 0$.

Term ① in Eq. 22 represents the additional knowledge of $F$ obtained from one extra training example beyond $S_{\mathcal{X}}^n, S_{\mathcal{Y}}^n$. It can be interpreted through the lens of the identifiability–agreement tradeoff. When $H\left(F \mid S_{\mathcal{X}}^n, S_{\mathcal{Y}}^n\right)$ is small, the reduction $H\left(F \mid S_{\mathcal{X}}^n, S_{\mathcal{Y}}^n\right) - H\left(F \mid S_{\mathcal{X}}^n, S_{\mathcal{Y}}^n, X, Y\right)$ is necessarily small, implying that strong identifiability of the ground-truth leads to a small lower bound for the optimal risk of a classification problem, which is consistent with the conventional sense. On the other hand, even if identifiability drops, i.e. $H\left(F \mid S_{\mathcal{X}}^n, S_{\mathcal{Y}}^n\right)$ is large, our bound suggests that it doesn't necessarily leads to a high difficulty of learning. As long as the classifiers in $\mathcal{F}$ share high agreement, then even if observing a new pair $(X, Y)$ will not significantly improve our knowledge about the ground-truth, the conditional entropy difference will still remain small, so as the lower bound of the optimal risk. Thus, compared to $H\left(F \mid S_{\mathcal{X}}^n, S_{\mathcal{Y}}^n\right)$, the conditional entropy difference can handle both ends of the identifiability-agreement balance, making it a more sufficient way to approximate the difficulty of learning a problem.

## 4.4 EXAMPLES

**Example 1.** We first revisit Example 1. Some of the essential quantities are given as follows,

$$R^* = \epsilon(1-\epsilon)^n / 2, \tag{23}$$
$$H\left(F \mid S_{\mathcal{X}}^n, S_{\mathcal{Y}}^n\right) = (1-\epsilon)^n, \tag{24}$$
$$H\left(F \mid S_{\mathcal{X}}^n, S_{\mathcal{Y}}^n\right) - H\left(F \mid S_{\mathcal{X}}^n, S_{\mathcal{Y}}^n, X, Y\right) = \epsilon(1-\epsilon)^n. \tag{25}$$

We begin by comparing Eq. 23 and Eq. 24. Notably, in this example, the agreement is exactly $1 - \epsilon$, making $\epsilon$ an explicit opposite measure of agreement. For $\epsilon \gg 0$ large enough, both the conditional entropy $H\left(F \mid S_{\mathcal{X}}^{n}, S_{\mathcal{Y}}^{n}\right) = (1 - \epsilon)^{n}$—which inversely reflects the ground-truth identifiability—and $R^{*}$ will decrease with $n$ and converge to $0$, despite the fact that the two hypotheses share low agreement. However, if $\epsilon$ is close to $0$, then for small $n < \infty$, $R^{*}$ will be close to $0$ while $H\left(F \mid S_{\mathcal{X}}^{n}, S_{\mathcal{Y}}^{n}\right)$ is nevertheless close to $1$, exhibiting a negative correlation. On the other hand, we notice from Eq. 25 that the conditional entropy reduction precisely characterizes the decay rate of the optimal risk with respect to $n$. Thus, the conditional entropy reduction $H\left(F \mid S_{\mathcal{X}}^{n}, S_{\mathcal{Y}}^{n}\right) - H\left(F \mid S_{\mathcal{X}}^{n}, S_{\mathcal{Y}}^{n}, X, Y\right)$ outperforms $H\left(F \mid S_{\mathcal{X}}^{n}, S_{\mathcal{Y}}^{n}\right)$ in characterizing the hardness of this problem for quantifying both identifiability and agreement simultaneously. By using Theorem 1, our lower bound $\mathrm{LB}_1$ for $R^{*}$ is given as follows.

$$\mathrm{LB}_1 = \frac{\epsilon^2 (1 - \epsilon)^{2n}}{4}. \tag{26}$$

Notice that our lower bound grows at square rate as the true optimal risk $R^{*}$. On the other hand, the bound of Chen et al. (2016) computed for Example 1 is $0$, which provide no informative approximation of the hardness of learning of the problem. An additional example that has a similar but different setup as this one can be found in Section B.1.

**Example 2.** Let $\mathcal{X} := \{a_1, \cdots, a_M\}$ be a finite input alphabet, and let $\mathcal{Y} = \{0, 1\}$. Let $\mathcal{F}^{\mu} := \{f : \mathcal{X} \to \mathcal{Y}\}$ be a set of all possible hard classifiers, We fix $\mathcal{E}_F^{\mu}$ as the uniform distribution on $\mathcal{F}$. For some $\epsilon \leq \frac{M-1}{M}$, we define the input distribution $\mu$ as follows,

$$\mu(x) = \begin{cases} 1 - \epsilon & x = a_1, \\ \dfrac{\epsilon}{M - 1} & x \in \{a_2, \cdots, a_M\}. \end{cases} \tag{27}$$

In this example we primarily consider the case when $n = 1$, and denote $S_{\mathcal{X}}^{n}, S_{\mathcal{Y}}^{n} = \{X_{\mathrm{S}}, Y_{\mathrm{S}}\}$, then the following metrics can be computed,

$$
\begin{aligned}
R^{*} &= \frac{1}{2}\left(1 - (1 - \epsilon)^2 - \frac{\epsilon^2}{M - 1}\right) \\
H\left(F \mid X_{\mathrm{S}}, Y_{\mathrm{S}}\right) &= M - 1 \\
H\left(F \mid X_{\mathrm{S}}, Y_{\mathrm{S}}\right) - H\left(F \mid X_{\mathrm{S}}, Y_{\mathrm{S}}, X, Y\right) &= 1 - (1 - \epsilon)^2 - \frac{\epsilon^2}{M - 1}.
\end{aligned} \tag{28}
$$

We notice that the conditional entropy difference changes as exactly the same rate as the optimal risk $R^{*}$ with respect to $\epsilon$. Since this example is also considering a binary classification problem, we can similarly obtain a lower bound whose rate with respect to $\epsilon$ is the square of that of the optimal risk, while the lower bound given by Chen et al. (2016) is $0$. Additionally, we also observe that the conditional entropy $H\left(F \mid X_{\mathrm{S}}, Y_{\mathrm{S}}\right)$ is a constant, implying that the identifiability of the problem remains unchanged regardless of $\epsilon$, hence it cannot adequately approximate the variable difficulty of learning of the problem with respect to the choice of $\epsilon$. The computation process of this example is provided in Section A.2. We have also extended the analysis of this example to arbitrary sample size $n$, which is discussed in Section B.2.

## 5 CONVERGENCE OF LOWER BOUND

As $n$ grows large, the additional knowledge that a single $(X, Y)$ can provide will decrease. On the other hand, it is a belief of a majority of the machine learning community that if a learner is provided with infinite training examples, it should be able to gain the maximum possible knowledge about the ground-truth, and its the optimal risk will have no room for further improvement. Thus, to fully characterize the properties of the conditional entropy reduction $H\left(F \mid S_{\mathcal{X}}^{n}, S_{\mathcal{Y}}^{n}\right) - H\left(F \mid S_{\mathcal{X}}^{n}, S_{\mathcal{Y}}^{n}, X, Y\right)$ under the influence of $n$, a natural question arises: does the conditional entropy reduction eventually converge to zero as $n$ increases? If so, at what rate is the convergence guaranteed? We will study this concern in this section.

Particularly, in this section we will consider a special case of classification problems. We suppose that $\mathcal{F}^{\mu}$ is finite. Let $\mathcal{M}(\epsilon, \mathcal{T}, d)$ denote the $\epsilon$-packing number of a set $\mathcal{T}$ under metric $d$. For any

$f_1, f_2 \in \mathcal{F}^\mu$, we define their $\mu$-TVD (Total Variation distance) between them as

$$d_{\text{TV}}^\mu(f_1, f_2) := \mathop{\mathbb{E}}_{x \sim \mu} d_{\text{TV}}(f_1(x), f_2(x)). \tag{29}$$

For any $f \in \mathcal{F}^\mu$ and $\epsilon > 0$, we define $\mathcal{F}_\epsilon^f := \{f' \in \mathcal{F}^\mu : d_{\text{TV}}^\mu(f, f') \leq \epsilon\}$. We make the following regularity assumptions.

Suppose that there are a sequence $(\epsilon_i)$ of positive numbers and a sequence $(\mathcal{F}_i)$ of subsets of $\mathcal{F}$, such that: $\lim_{i \to \infty} \epsilon_i = 0$, $\lim_{i \to \infty} i\epsilon_i^2 = \infty$, $\lim_{i \to \infty} \sum_{j=1}^{i} \exp\left(-Bj\epsilon_j^2\right) < \infty$ for all $B > 0$, $\log \mathcal{M}\left(\epsilon_i, \mathcal{F}_i, d_{\text{TV}}^\mu\right) \leq i\epsilon_i^2$, and for some constant $C > 0$: $\mathcal{E}_F^\mu\left(\mathcal{F} \backslash \mathcal{F}_i\right) \leq \exp\left(-i\epsilon_i^2(C+4)\right)$ and $\min_{f \in \mathcal{F}^\mu} \mathcal{E}_F^\mu(f) \geq \exp\left(-i\epsilon_i^2 C\right)$.

**Theorem 2.** *For some constant $B > 0$, we define*

$$n^* := \min\left\{N \in \mathbb{N}_{>0} : \frac{1 - \exp\left(-Bn'\epsilon_{n'}^2\right)}{\max\limits_{f \in \mathcal{F}^\mu} \left|\mathcal{F}_{M\epsilon_{n'}}^f\right|} \geq \frac{\exp\left(-Bn'\epsilon_{n'}^2\right)}{|\mathcal{F}| - \max\limits_{f \in \mathcal{F}^\mu} \left|\mathcal{F}_{M\epsilon_{n'}}^f\right|}, \quad \forall n' > N\right\}, \tag{30}$$

*and for all $n > n^*$, the conditional entropy reduction is upper bounded as follows,*

$$H\left(F \mid S_\mathcal{X}^n, S_\mathcal{Y}^n\right) - H\left(F \mid S_\mathcal{X}^n, S_\mathcal{Y}, X, Y\right) \leq \log \max_{f \in \mathcal{F}^\mu} \left|\mathcal{F}_{M\epsilon_n}^f\right| + \mathcal{O}\left(\exp\left(-Bn\epsilon_n^2\right) \log |\mathcal{F}^\mu|\right). \tag{31}$$

The proof of Theorem 2 can be found in Section G. From the theorem, the convergence of the conditional entropy reduction is deterministically dominated by $\exp\left(-Bn\epsilon_n^2\right)$. To interpret this term, consider the following special case. Suppose that we choose $\epsilon_i$ as $\alpha i^{-\beta}$ for some $\alpha \in (0, \infty)$ and $\beta \in (0, \frac{1}{2})$. It can be verified that by carefully choosing the value of $\alpha$, the regularity assumptions will all be made to hold true. In such cases, the term $\exp\left(-Bn\epsilon_n^2\right)$ will decay at rate $\exp(-i^{1-2\beta}B\alpha)$.

Then, according to Theorem 2, the convergence rate of the second term in the RHS of Eq. 31 is dominated by $\exp\left(-i^{1-2\beta}B'\right)$ for some constant $B' > 0$. If we further choose $\beta$ to be extremely close to $0$, then the second term will converge to $0$ approximately at exponential rate $\exp\left(-iB'\right)$, which aligns with what we observed in Example 1. This result reinforces the consistency between our theory and practice. On the other hand, by choosing $\epsilon_i$ in the aforementioned manner, $\max\limits_{f \in \mathcal{F}^\mu} \left|\mathcal{F}_{M\epsilon_n}^f\right|$ is also decreasing with $n$. Thus, the first term on the RHS of Eq. 31 will also keep reducing to $1$ with $n$ growing large enough.

# 6 CONCLUSION AND LIMITATION

In this work, we have made two main contributions. Firstly, we have derived lower bounds on the optimal risk under average notions of risk measures for classification problems. Our bound is implicitly governed by conditional entropy reduction

$$H(F \mid S_\mathcal{X}^n, S_\mathcal{Y}^n) - H(F \mid S_\mathcal{X}^n, S_\mathcal{Y}^n, X, Y), \tag{32}$$

which uncovers an implicit tradeoff between identifiability and agreement, two quantities that jointly govern the performance of the optimal learner in a classification problem. Using some toy examples, we have demonstrated that our bound is stronger than existing ones that are also applicable in classification problems. Secondly, we have proved the asymptotic behavior of our derived bound with respect to the size of the training sample. We show that with careful selection of certain hyperparameters, our bound will converge to $0$ at an approximately exponential rate.

There are also limitations in our study. While the problem formulation is general, it still require more modeling techniques to be applied to arbitrary practical scenarios. We look forward to investigating whether similar bounds can also be derived in other problems like regression, unsupervised problems, etc., and whether the identifiability-agreement tradeoff can be extended to general cases. Moreover, the analysis of the convergence behavior and the examples focuses primarily on classification problems with finite hypothesis classes or hard classifiers only, whereas extending it to more general paradigms remains an open challenge.

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

## A   COMPUTATIONS OF EXAMPLES

### A.1   EXAMPLE 1

Let $\Gamma := [1/2 - \epsilon/2, 1/2 + \epsilon/2]$ and $\Gamma^c := [0,1]\backslash\Gamma$. When $S_{\mathcal{X}}^n$ contains at least one observation in $\Gamma$, the optimal learner achieves 0 risk. If no such sample point is included, then the error is $1/2$ whenever the test point $X$ falls within $\Gamma$, and 0 otherwise. Therefore, the expected optimal risk is:

$$R^* = \Pr(\exists x \in S_{\mathcal{X}}^n : x \in \Gamma) \times 0$$
$$+ \Pr(\forall x \in S_{\mathcal{X}}^n : x \in \Gamma^c) \times \left( \Pr(X \in \Gamma^c) \times 0 + \Pr(X \in \Gamma) \times \frac{1}{2} \right)$$
$$= n\epsilon \times 0 + (1 - \epsilon)^n \times \epsilon \times \frac{1}{2}$$
$$= \frac{\epsilon(1 - \epsilon)^n}{2}$$

Similarly, when $s_{\mathcal{X}}^n$ includes at least one point in $\Gamma$, the posterior $\mathcal{E}_{F|S_{\mathcal{X}}^n, S_{\mathcal{Y}}^n}^\mu(\cdot \mid s_{\mathcal{X}}^n, s_{\mathcal{Y}}^n)$ is one-hot, and the entropy of this distribution is zero; Otherwise, the posterior is uniform over $\{f_a, f_b\}$, with entropy equal to one. Hence, the corresponding conditional entropy can be expressed as:

$$H(F \mid S_{\mathcal{X}}^n, S_{\mathcal{Y}}^n) = \Pr(\exists x \in S_{\mathcal{X}}^n : x \in \Gamma) \times 0 + \Pr(\forall x \in S_{\mathcal{X}}^n : x \in \Gamma^c) \times 1$$
$$= (1 - \epsilon)^n$$

which directly implies $H(F \mid S_{\mathcal{X}}^n, S_{\mathcal{Y}}^n, X, Y) = (1 - \epsilon)^{n+1}$ and thus we have that $H(F \mid S_{\mathcal{X}}^n, S_{\mathcal{Y}}^n) - H(F \mid S_{\mathcal{X}}^n, S_{\mathcal{Y}}^n, X, Y) = \epsilon^n(1 - \epsilon)$.

### A.2   EXAMPLE 2

For any observation $X_S = a_i$, the error of the optimal learner at the test point $X = a_i$ is zero, while at any other point $X \in \mathcal{X}\backslash\{a_i\}$, it is $1/2$:

$$R^* = \sum_{i=1}^{M} \Pr(X_S = a_i) \left( \Pr(X = a_i) \times 0 + \Pr(X \in \mathcal{X}\backslash\{a_i\}) \times \frac{1}{2} \right)$$
$$= \Pr(X_S = a_1) \left( \Pr(X = a_1) \times 0 + \Pr(X \in \mathcal{X}\backslash\{a_1\}) \times \frac{1}{2} \right)$$
$$+ (M - 1)\Pr(X_S = a_2) \left( \Pr(X = a_2) \times 0 + \Pr(X \in \mathcal{X}\backslash\{a_2\}) \times \frac{1}{2} \right)$$
$$= (1 - \epsilon) \times \frac{\epsilon}{2} + (M - 1) \times \frac{\epsilon}{M - 1} \times \frac{(1 - \epsilon) + \epsilon - \frac{\epsilon}{M-1}}{2}$$
$$= \frac{1}{2} \left( 1 - (1 - \epsilon)^2 - \frac{\epsilon^2}{M - 1} \right)$$

Whenever the observation is $X_S = a_i$ for any $i \in [M]$, the uncertainty of the classifier is confined to the $2^{M-1}$ hypotheses associated with the points in $\mathcal{X}\backslash\{a_i\}$. As the prior $\mathcal{E}_F$ is uniform, the posterior remains uniform over this set, which gives:

$$H(F \mid X_S, Y_S) = \mathbb{E}_{a \sim \mu}[H(F \mid X_S = a, Y_S)]$$
$$= \mathbb{E}_{a \sim \mu}[2^{M-1} \times 2^{-(M-1)} \log 2^{M-1}]$$
$$= M - 1.$$

However, when the sample size is $n = 2$—for simplicity let $(X_1, Y_1)$ and $(X_2, Y_2)$ denote the two data pairs— as the sampling process is i.i.d., it is possible that $X_1 = X_2$. Accordingly, three cases arise:

1. $X_1 = X_2 = a_1$, with probability $\Pr(X_1 = X_2 = a_1) = (1 - \epsilon)^2$,

2. $X_1 = X_2 = a_j$ for some $j \neq 1$, with probability $\Pr(X_1 = X_2 = a_j) = \frac{\epsilon^2}{M-1}$,

3. $X_1 \neq X_2$, with probability $\Pr(X_1 \neq X_2) = 1 - (1 - \epsilon)^2 - \frac{\epsilon^2}{M-1}$,

In the first two cases, the conditional entropy $H(F \mid \text{Case 1 or 2}) = 2^{M-1} \times 2^{-(M-1)} \log 2^{M-1} = M - 1$ does not change. In the last case, the conditional entropy decreases to $H(F \mid \text{Case 3}) = 2^{M-2} \times 2^{-(M-2)} \log 2^{M-2} = M - 2$. Thus,

$$
\begin{aligned}
H(F \mid X_{\mathrm{S}}, Y_{\mathrm{S}}, X, Y) &= H(F \mid X_1, Y_1, X_2, Y_2) \\
&= \Pr(X_1 = X_2) \times H(F \mid \text{Case 1 or 2}) + \Pr(X_1 \neq X_2) H(F \mid \text{Case 3}) \\
&= \left( (1 - \epsilon)^2 + \frac{\epsilon^2}{M-1} \right) \times (M - 1) + \left( 1 - (1 - \epsilon)^2 - \frac{\epsilon^2}{M-1} \right) \times (M - 2) \\
&= (M - 1) - 2\epsilon + 2\epsilon^2 - \frac{(M-2)\epsilon^2}{M-1}.
\end{aligned}
$$

Then,

$$
\begin{aligned}
H(F \mid X_{\mathrm{S}}, Y_{\mathrm{S}}) - H(F \mid X_{\mathrm{S}}, Y_{\mathrm{S}}, X, Y) &= M - 1 - \left( (M - 1) - 2\epsilon + 2\epsilon^2 - \frac{(M-2)\epsilon^2}{M-1} \right) \\
&= 2\epsilon - 2\epsilon^2 + \frac{(M-1)\epsilon^2 - \epsilon^2}{M-1} \\
&= 2\epsilon - \epsilon^2 - \frac{\epsilon^2}{M-1} \\
&= 1 - (1 - \epsilon)^2 - \frac{\epsilon^2}{M-1}
\end{aligned}
$$

## B  ADDITIONAL EXAMPLES

### B.1  EXAMPLE 3

Consider the same setup as in Example 1, except that $f_a$ is redefined as follows:

$$
f_a(x) = \begin{cases} +1 & x \in [0, \frac{1}{2} - \frac{1}{2}\epsilon), \\ -1 & x \in [\frac{1}{2} - \frac{1}{2}\epsilon, 1]. \end{cases} \tag{33}
$$

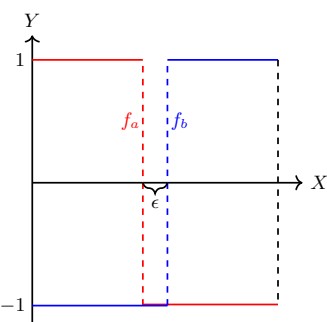

Figure 3: Example 3

Figure 3 provides an illustration of this example. The following results can be verified,

$$
R^* = \epsilon^n (1 - \epsilon)/2, \tag{34}
$$

$$
H\left( F \mid S_{\mathcal{X}}^n, S_{\mathcal{Y}}^n \right) = \epsilon^n, \tag{35}
$$

$$
H\left( F \mid S_{\mathcal{X}}^n, S_{\mathcal{Y}}^n \right) - H\left( F \mid S_{\mathcal{X}}^n, S_{\mathcal{Y}}^n, X, Y \right) = \epsilon^n (1 - \epsilon). \tag{36}
$$

An inverse form of the trade-off identified in Example 1 is observed here, which is also explicitly quantified by the conditional entropy difference.

Same as in Example 1, according to Theorem 1, our lower bound $\text{LB}_3$ for this case grows at square rate as the true optimal risk $R^*$, which is given as follows.

$$\text{LB}_3 = \frac{\epsilon^{2n}(1-\epsilon)^2}{4}, \tag{37}$$

while the bound of Chen et al. (2016) is still constantly 0.

The computations of the measures in this example follow the same logic with that of Example 1. In contrast to Example 1, in this setting, if $S_{\mathcal{X}}^n$ contains at least one point in $\Gamma^c$, the optimal learner achieves 0 risk. If all observed points fall within $\Gamma$, then the error is $1/2$ whenever the test point $X$ lies in $\Gamma$, and 0 otherwise. Accordingly, the expected optimal risk is:

$$\begin{aligned}
R_2^* &= \Pr(\exists x \in S_{\mathcal{X}}^n : x \in \Gamma^c) \times 0 \\
&\quad + \Pr(\forall x \in S_{\mathcal{X}}^n : x \in \Gamma) \times \left( \Pr(X \in \Gamma) \times 0 + \Pr(X \in \Gamma^c) \times \frac{1}{2} \right) \\
&= n(1-\epsilon) \times 0 + \epsilon^n \times (1-\epsilon) \times \frac{1}{2} \\
&= \frac{\epsilon^n(1-\epsilon)}{2}
\end{aligned}$$

And the corresponding conditional entropy are:

$$\begin{aligned}
H(F \mid S_{\mathcal{X}}^n, S_{\mathcal{Y}}^n) &= \Pr(\exists x \in S_{\mathcal{X}}^n : x \in \Gamma^c) \times 0 + \Pr(\forall x \in S_{\mathcal{X}}^n : x \in \Gamma) \times 1 = \epsilon^n \\
&\Rightarrow H(F \mid S_{\mathcal{X}}^n, S_{\mathcal{Y}}^n, X, Y) = \epsilon^{n+1} \\
&\Rightarrow H(F \mid S_{\mathcal{X}}^n, S_{\mathcal{Y}}^n) - H(F \mid S_{\mathcal{X}}^n, S_{\mathcal{Y}}^n, X, Y) = \epsilon^n(1-\epsilon)
\end{aligned}$$

### B.2 EXAMPLE 4

This example follows the same setting as Example 2, but we will consider large sample size $n \geq M$. Then, we can compute the corresponding metrics as follows,

$$\begin{aligned}
R^* &\leq \mathcal{O}\left( \sum_{k=1}^{M} g(k) \right), \\
H\left(F \mid S_{\mathcal{X}}^n, S_{\mathcal{Y}}^n\right) - H\left(F \mid S_{\mathcal{X}}^n, S_{\mathcal{Y}}^n, X, Y\right) &= \mathcal{O}\left( M \sum_{k=1}^{M} g(k) \right),
\end{aligned} \tag{38}$$

where

$$g(k) = \left( \frac{(k-1)\epsilon}{M-1} + 1 - \epsilon \right)^n \frac{M - 1 - k\epsilon + \epsilon}{M-1}. \tag{39}$$

We observe that the conditional entropy difference exhibits the same rate of growth with respect to $n$ as the optimal risk (up to a constant factor of $M$ which is assumed to be finite). Thus, we may conclude that the conditional entropy difference can still provide a strong approximation of the optimal risk.

Below we will provide a draft of the computations for this example. Notice that one essential quantity that needs to be analyzed in this example is the probability that $k$ distinct symbols are observed in $n$ i.i.d. sampled input points for each $k = 1, \cdots, M$. To do this, we will need to apply Sanov's Theorem (Sanov, 1958), which we first briefly introduce as follows.

In addition to the setup of this example, we will let $\mathbf{x}$ denote $(x_1, \cdots, x_n)$. For any $a \in \mathcal{X}$ and $\mathbf{x}$, we denote by $N(a \mid \mathbf{x})$ the number of occurrences of $a$ in $\mathbf{x}$. We define the type $P_{\mathbf{x}}$ to be a function mapping $\mathcal{X}$ to the set $\{0, \frac{1}{n}, \frac{2}{n}, \cdots, 1\}$ defined by $P_{\mathbf{x}}(a) = \frac{N(a|\mathbf{x})}{n}$ for any $a \in \mathcal{X}$. Such a type may also be called a type with denominator $n$. We will denote by $\mathcal{P}_n$ the set of all types with denominator $n$. Notice that $\mathcal{P}_n \subset \Delta(\mathcal{X})$. Then, Sanov's Theorem (Sanov, 1958) states as follows.

**Theorem 3** (Sanov's Theorem). *Let $E \subset \Delta(\mathcal{X})$. Let $\mathbf{p}^* = \arg\min_{\mathbf{p} \in E} d_{\text{KL}}(\mathbf{p} \| \mu)$. Then,*

- $\mu^n(E \cap \mathcal{P}_n) \leq (n+1)^{|\mathcal{X}|} 2^{-n d_{\text{KL}}(\mathbf{p}^* \| \mu)}.$

- *If, in addition, $E$ is the closure of ts interior, then*

$$\mu^n(E \cap \mathcal{P}_n) \doteq 2^{-n d_{\mathrm{KL}}(\mathbf{p}^* \| \mu)}. \tag{40}$$

Given $k$, define $E = \{\mathbf{p} \in \Delta(\mathcal{X}) : |\Omega(\mathbf{p})| = k\}$. Then the probability of observing $k$ distinct symbols in $n$ input points i.i.d. sampled from $\mu$ can be equivalently written as $\mu^n(E \cap \mathcal{P}_n)$. To derive the closed-form solution of this probability, it suffices to solve the optimization program defined as follows,

$$\begin{aligned} \min_{\mathbf{p} \in E} \quad & d_{\mathrm{KL}}(\mathbf{p} \| \mu), \\ \text{subject to} \quad & |\Omega(\mathbf{p})| = k. \end{aligned} \tag{41}$$

We may divide the program into two cases. Case 1: Suppose that $\mathbf{p}_1^*$ has positive probability on $a_1$ (recall that $\mu(a_1) = 1 - \epsilon$). Then the optimization program can be reformulated as follows,

$$\begin{aligned} \min_{\mathbf{p} \in \mathbb{R}^k} \quad & p_1 \log \frac{p_1}{1 - \epsilon} + \sum_{i=2}^{k} p_i \log \frac{p_i}{\epsilon'}, \\ \text{subject to} \quad & -p_i < 0 \quad \forall i = 1, \cdots, k, \\ & \mathbf{p}^\top I - 1 = 0, \end{aligned} \tag{42}$$

where $\epsilon' = \frac{\epsilon}{M-1}$. It can be verified that this is a convex optimization program. Then by using KKT conditions and Lagrange Multiplier Method, we can solve for the program and have that $d_{\mathrm{KL}}(\mathbf{p}_1^* \| \mu) = \log \dfrac{M-1}{k\epsilon + M(1-\epsilon) - 1}$.

Case 2: Suppose that $\mathbf{p}_2^*(a_1) = 0$. Then the program can equivalently be written as follows,

$$\begin{aligned} \min_{\mathbf{p} \in \mathbb{R}^k} \quad & \sum_{i=1}^{k} p_i \log \frac{p_i}{\epsilon'}, \\ \text{subject to} \quad & -p_i < 0 \quad \forall i = 1, \cdots, k, \\ & \mathbf{p}^\top I - 1 = 0. \end{aligned} \tag{43}$$

Similarly, we can solve for the program and have that $d_{\mathrm{KL}}(\mathbf{p}_1^* \| \mu) = \log \dfrac{M-1}{k\epsilon}$. Since $\log \dfrac{M-1}{k\epsilon + M(1-\epsilon) - 1} \leq \log \dfrac{M-1}{k\epsilon}$, the final solution will be $d_{\mathrm{KL}}(\mathbf{p}^* \| \mu) = \log \dfrac{M-1}{k\epsilon + M(1-\epsilon) - 1}$. Thereby, we have derived the closed-form solution for the probability of observing $k$ distinct symbols in $n$ i.i.d. sampled input points, and the rest of the computation of this example will be solved.

## C   PROOF OF LEMMA 1

Fact 1: For any discrete probability measures $P$ and $Q$, it holds that $\mathbb{E}_{X \sim P} \mathbb{E}_{X' \sim Q} \mathbb{1}\{X \neq X'\} = 1 - \langle P, Q \rangle$.

Proof of Fact 1:

$$\begin{aligned} \mathbb{E}_{X \sim P} \mathbb{E}_{X' \sim Q} \mathbb{1}\{X \neq X'\} &= \sum_x P(x) \sum_{x'} Q(x') \mathbb{1}\{X \neq X'\} \\ &= \sum_x P(x)(1 - Q(x)) \\ &= \sum_x P(x) - \sum_x P(x)Q(x) \\ &= 1 - \langle P, Q \rangle \end{aligned}$$

Fact 2: Let $\nu$ be a joint distribution of random variables $A, B$, then for any function $f$:

$$\mathbb{E}_{a \sim \nu_A} f(a) = \mathbb{E}_{a \sim \nu_A} \mathbb{E}_{b \sim \nu_{B|A=a}} \mathbb{E}_{a' \sim \nu_{A|B=b}} f(a'). \tag{44}$$

Proof of Fact 2: fix $b$, the relation is given by $\mathbb{E}_{a'\sim\nu_{A|B=b}}f(a') = \mathbb{E}\left[f(A) \mid B = b\right]$. Then, the RHS of Eq. 44 equals to

$$\begin{aligned}
\mathbb{E}_{A,B,A'}f(A') &= \mathbb{E}_{A,B}\mathbb{E}[f(A') \mid B] \\
&= \mathbb{E}_{A,B}\mathbb{E}[f(A) \mid B] \\
&= \mathbb{E}_A f(A).
\end{aligned}$$

We now prove Lemma 1 using above facts:

$$R(\mathscr{A};\mu,\mathcal{E}_F) \overset{(a)}{=} \mathbb{E}_{x\sim\mu} \underset{\substack{f\sim\mathbb{P}_F \\ s_{\mathcal{X}}^n\sim\mathbb{P}_{S_{\mathcal{X}}^n} \\ s_{\mathcal{Y}}^n\sim\mathbb{P}_{S_{\mathcal{Y}}^n|S_{\mathcal{X}}^n=s_{\mathcal{X}}^n,F=f}}{\mathbb{E}} \underset{\substack{f'\sim\mathbb{P}_{F|S_{\mathcal{X}}^n=s_{\mathcal{X}}^n,S_{\mathcal{Y}}^n=s_{\mathcal{Y}}^n} \\ \hat{f}\sim\mathscr{A}(s_{\mathcal{X}}^n,s_{\mathcal{Y}}^n)}}{\mathbb{E}} \left[\underset{\substack{y\sim f'(x) \\ \hat{y}\sim\hat{f}(x)}}{\mathbb{E}} \mathbb{1}\{y\neq\hat{y}\}\right]$$

$$\overset{(b)}{=} \mathbb{E}_{x\sim\mu}\mathbb{E}_{s_{\mathcal{X}}^n,s_{\mathcal{Y}}^n\sim\mathbb{P}_{S_{\mathcal{X}}^n,S_{\mathcal{Y}}^n}}\mathbb{E}_{f'\sim\mathcal{E}_{F|S_{\mathcal{X}}^n,S_{\mathcal{Y}}^n}^\mu(\cdot|s_{\mathcal{X}}^n,s_{\mathcal{Y}}^n),\hat{f}\sim\mathscr{A}(s_{\mathcal{X}}^n,s_{\mathcal{Y}}^n)}\left[1 - \left\langle f'(x),\hat{f}(x)\right\rangle\right]$$

$$\overset{(c)}{=} \mathbb{E}_{x\sim\mu}\mathbb{E}_{s_{\mathcal{X}}^n,s_{\mathcal{Y}}^n\sim\mathbb{P}_{S_{\mathcal{X}}^n,S_{\mathcal{Y}}^n}}\left[1 - \left\langle \mathbb{E}_{f'\sim\mathcal{E}_{F|S_{\mathcal{X}}^n,S_{\mathcal{Y}}^n}^\mu(\cdot|s_{\mathcal{X}}^n,s_{\mathcal{Y}}^n)}f'(x), \mathbb{E}_{\hat{f}\sim\mathscr{A}(s_{\mathcal{X}}^n,s_{\mathcal{Y}}^n)}\hat{f}(x)\right\rangle\right],$$

where step $(a)$ follows from Fact 1, step $(b)$ follows from Fact 2, and step $(c)$ uses bilinearity of the inner product and the conditional independence between $f'$ and $\hat{f}$ given the sample $(s_{\mathcal{X}}^n, s_{\mathcal{Y}}^n)$. $\quad\square$

## D  PROOF OF PROPOSITION 1

**Lemma 3.** *Let $\mathbf{u} = [u_1,\cdots,u_m]^\top$ and $\mathbf{v} = [v_1,\cdots,v_m]^\top$ be 2 finite-dimensional vectors that satisfy the following conditions:*

- $u_i, v_i \geq 0$ *for all* $i = 1,\cdots,m$,
- $\sum_i u_i = \sum_i v_i = 1$,

*then for any fixed $\mathbf{u}$, the inner product $\langle\mathbf{u},\mathbf{v}\rangle$ is maximized with respect to $\mathbf{v}$ when $\mathbf{v} = \mathbf{e}^{(i^*)}$ for some $i^* \in \underset{i}{\arg\max}\, u_i$, where $\mathbf{e}^{(i)}$ denotes the standard basis vector with a 1 at position $i$.*

**Proof of Lemma 3:**
By definition:

$$\langle\mathbf{u},\mathbf{e}^{(i^*)}\rangle = u_{i^*} \times 1 = u_{i^*}.$$

Since $u_j \leq u_{i^*}$ for any $j \in [m]$, it follows that for any $\mathbf{v}$:

$$\langle\mathbf{u},\mathbf{v}\rangle = \sum_{i=1}^m u_i v_i \leq \sum_{i=1}^m u_{i^*}v_i \leq u_{i^*}\sum_{i=1}^m v_i = u_{i^*} = \langle\mathbf{u},\mathbf{e}^{(i^*)}\rangle.$$

$\square$

**Proof of Proposition 1:**
By Lemma 1, the optimal learner is

$$\begin{aligned}
\mathscr{A}^* &= \underset{\mathscr{A}}{\arg\min}\, R\left(\mathscr{A};\mu,\mathcal{E}_F\right) \\
&= \underset{\mathscr{A}}{\arg\max}\, \mathbb{E}_{x\sim\mu}\mathbb{E}_{s_{\mathcal{X}}^n,s_{\mathcal{Y}}^n\sim\mathbb{P}_{S_{\mathcal{X}}^n,S_{\mathcal{Y}}^n}}\left\langle \mathbb{E}_{f'\sim\mathcal{E}_{F|S_{\mathcal{X}}^n,S_{\mathcal{Y}}^n}^\mu(\cdot|s_{\mathcal{X}}^n,s_{\mathcal{Y}}^n)}f'(x), \mathbb{E}_{\hat{f}\sim\mathscr{A}(s_{\mathcal{X}}^n,s_{\mathcal{Y}}^n)}\hat{f}(x)\right\rangle \\
&:= \underset{v}{\arg\max}\, \mathbb{E}_X\mathbb{E}_S\left\langle u_{Y|X,S}, v_{Y|X,S}\right\rangle
\end{aligned}$$

The second line can be interpreted as the expected inner product between two discrete conditional distributions, denoted by $u$ and $v$ for clarity. Lemma 3 states that for any $x$ and $s$, $\max_v \left\langle u_{Y|X,S}(\cdot \mid x,s), v_{Y|X,S}(\cdot \mid x,s)\right\rangle$ is achieved when $v_{Y|X,S}(\cdot \mid x,s)$ is the one-hot distribution concentrated on $\arg\max_y u_{Y|X,S}(y \mid x,s)$, and the corresponding maximum value is $\max_y u_{Y|X,S}(y \mid x,s)$.

Substituting $u, v$ back into our notation, we obtain that for any given sample $s_{\mathcal{X}}^n, s_{\mathcal{Y}}^n$, the optimal learner deterministically returns a following hard classifier: for any input $x$, it outputs the label:

$$y^* := \arg\max_y \bar{f}_{s_{\mathcal{X}}^n, s_{\mathcal{Y}}^n}(y \mid x)$$

with probability 1. $\qquad\square$

## E   PROOF OF LEMMA 2

Fano's inequality typically applies to a Markov chain of the form

$$\widetilde{Y} - \widetilde{X} - \hat{\widetilde{Y}}, \tag{45}$$

where $\widetilde{Y}$ and $\hat{\widetilde{Y}}$ take values from a finite alphabet of size $K$. Let

$$p_{\mathrm{e}} := \Pr\left\{\widetilde{Y} \neq \hat{\widetilde{Y}}\right\} \tag{46}$$

denotes the probability of error. Fano's inequality states that

$$H(\widetilde{Y} \mid \widetilde{X}) \leq \mathscr{H}_{\mathrm{b}}(p_{\mathrm{e}}) + p_{\mathrm{e}} \log(K - 1). \tag{47}$$

In our setting, we instantiate this inequality by identifying

$$\widetilde{Y} = Y, \quad \hat{\widetilde{Y}} = \hat{Y}, \quad \widetilde{X} = (S_{\mathcal{X}}^n, S_{\mathcal{Y}}^n, \hat{F}, X). \tag{48}$$

With this substitution, the Markov structure required by Fano's inequality holds, and applying the inequality directly yields Eq. 20 in our paper. $\qquad\square$

**Remark 2.** *Our contribution is to make this Bayesian perspective of standard Fano's inequality concrete specifically for classification problems, by modeling labeling functions as random parameters that generates labels for input data drawn from an input distribution $\mu$. Prior works such as Chen et al. (2016) do not formulate a model as concrete as ours on classification problems and therefore do not obtain our clean, explicit interpretation. This concreteness also enables our insight into the identifiability–agreement trade-off.*

## F   PROOF OF THEOREM 1

We first decompose the conditional entropy $H\left(Y \mid S_{\mathcal{X}}^n, S_{\mathcal{Y}}^n, \hat{F}, X\right)$ which appeared in Fano's Inequality Eq. 20 as follows,

$$
\begin{aligned}
H\left(Y \mid S_{\mathcal{X}}^n, S_{\mathcal{Y}}^n, \hat{F}, X\right) &\overset{(a)}{=} H\left(Y \mid S_{\mathcal{X}}^n, S_{\mathcal{Y}}^n, X\right) \\
&\overset{(b)}{=} I\left(F; Y \mid S_{\mathcal{X}}^n, S_{\mathcal{Y}}^n, X\right) + H\left(Y \mid S_{\mathcal{X}}^n, S_{\mathcal{Y}}^n, F, X\right) \\
&\overset{(c)}{=} I\left(F; Y \mid S_{\mathcal{X}}^n, S_{\mathcal{Y}}^n, X\right) + H\left(Y \mid F, X\right) \\
&\overset{(d)}{=} I\left(F; (X, Y) \mid S_{\mathcal{X}}^n, S_{\mathcal{Y}}^n\right) - I\left(F; X \mid S_{\mathcal{X}}^n, S_{\mathcal{Y}}^n\right) + H\left(Y \mid F, X\right) \\
&\overset{(e)}{=} I\left(F; (X, Y) \mid S_{\mathcal{X}}^n, S_{\mathcal{Y}}^n\right) + H\left(Y \mid F, X\right) \\
&\overset{(f)}{=} H\left(F \mid S_{\mathcal{X}}^n, S_{\mathcal{Y}}^n\right) - H\left(F \mid S_{\mathcal{X}}^n, S_{\mathcal{Y}}^n, X, Y\right) + H\left(Y \mid F, X\right),
\end{aligned}
\tag{49}
$$

where $(a)$ holds since $Y \perp \hat{F} \mid \left(S_{\mathcal{X}}^n, S_{\mathcal{Y}}^n, X\right)$, $(b)$ and $(f)$ follows from the definition of mutual information, $(c)$ holds since $Y \perp \left(S_{\mathcal{X}}^n, S_{\mathcal{Y}}^n\right) \mid (F, X)$, $(b)$, $(d)$ follows from the chain rule of mutual information, $(e)$ follows from the fact that $F \perp X$. Denote the RHS of Eq. 49 by $\Lambda_{\mu, \mathcal{E}_F}$.

When $K > 2$:
By Fano's Inequality Eq. 20, we have that

$$
\begin{aligned}
\Lambda_{\mu, \mathcal{E}_F} &\leq \mathscr{H}_{\mathrm{b}}\left(R\left(\mathscr{A}; \mu, \mathcal{E}_F\right)\right) + R\left(\mathscr{A}; \mu, \mathcal{E}_F\right) \log(K - 1) \\
&\leq 1 + R\left(\mathscr{A}; \mu, \mathcal{E}_F\right) \log(K - 1),
\end{aligned}
\tag{50}
$$

where the second inequality holds since binary entropy is upper bounded by 1. Thus, we have the following lower bound on the overall risk,

$$R\left(\mathscr{A};\mu,\mathcal{E}_F\right) \geq \frac{\Lambda_{\mu,\mathcal{E}_F} - 1}{\log(K-1)}. \tag{51}$$

When $K = 2$:

Denote $R\left(\mathscr{A};\mu,\mathcal{E}_F\right)$ by $R$ for simplicity. In this case, Fano's Inequality Eq. 20 can be written as follows,

$$\begin{aligned}
\Lambda_{\mu,\mathcal{E}_F} &\leq \mathscr{H}_{\mathrm{b}}(R) + R\log(K-1) \\
&= \mathscr{H}_{\mathrm{b}}(R) \\
&= R\log\frac{1}{R} + (1-R)\log\frac{1}{1-R} \\
&\leq 2\sqrt{R\left(1-R\right)},
\end{aligned} \tag{52}$$

which leads to the following lower bound on $R$,

$$R(\mathscr{A};\mu,\mathcal{E}_F) \geq \frac{\Lambda_{\mu,\mathcal{E}_F}^2}{4}. \tag{53}$$

The proof is thereby completed. $\qquad\square$

## G   PROOF OF THEOREM 2

For any ground-truth classifier $f \sim \mathcal{E}_F^\mu$, with mild abuse of notations, we denote by $S_{\mathcal{Y},f}^n \sim f(S_{\mathcal{X}}^n)$ to highlight the dependency of its sampling process with respect to $f$. Also, we will denote by $H^{\mathrm{D}}\left(F \mid s_{\mathcal{X}}^n, s_{\mathcal{Y}}^n\right)$ the disintegrated conditional entropy for $S_{\mathcal{X}}^n, S_{\mathcal{Y}}^n = s_{\mathcal{X}}^n, s_{\mathcal{Y}}^n$. Thereby, the following equalities can be verified.

$$\begin{aligned}
H\left(F \mid S_{\mathcal{X}}^n, S_{\mathcal{Y}}^n\right) &= \mathop{\mathbb{E}}_{s_{\mathcal{X}}^n, s_{\mathcal{Y}}^n \sim \mathbb{P}_{S_{\mathcal{X}}^n, S_{\mathcal{Y}}^n}} H^{\mathrm{D}}\left(F \mid s_{\mathcal{X}}^n, s_{\mathcal{Y}}^n\right) \\
&= \mathop{\mathbb{E}}_{f\sim\mathcal{E}_F^\mu} \mathop{\mathbb{E}}_{\substack{s_{\mathcal{X}}^n \sim \mu^n \\ s_{\mathcal{Y},f}^n \sim f(s_{\mathcal{X}}^n)}} H^{\mathrm{D}}\left(F \mid s_{\mathcal{X}}^n, s_{\mathcal{Y},f}^n\right) \\
&= \mathop{\mathbb{E}}_{f\sim\mathcal{E}_F^\mu} H\left(F \mid S_{\mathcal{X}}^n, S_{\mathcal{Y},f}^n\right).
\end{aligned} \tag{54}$$

Then, using the main theorem of Ghosal et al. (2000), there exist constants $B_1, B_2 > 0$ such that the following holds for any ground-truth $f \in \mathcal{F}^\mu$,

$$\mathop{\Pr}_{\substack{s_{\mathcal{X}}^n \sim \mu^n \\ s_{\mathcal{Y},f}^n \sim f(s_{\mathcal{X}}^n)}} \left\{ \mathcal{E}_{F|S_{\mathcal{X}}^n, S_{\mathcal{Y}}^n}^\mu\left(f' : d_{\mathrm{TV}}^\mu(f', f) > M\epsilon_n \mid s_{\mathcal{X}}^n, s_{\mathcal{Y},f}^n\right) \leq \exp\left(-B_2 n\epsilon_n^2\right) \right\} \geq 1 - \exp\left(-B_1 n\epsilon_n^2\right), \tag{55}$$

where $M \geq \sqrt{\dfrac{C+4}{C_1}}$   for some $C_1 > 0$.

Define

$$n^* := \min\left\{ N \in \mathbb{N}_{>0} : \frac{1 - \exp\left(-B_2 n'\epsilon_{n'}^2\right)}{\max_{f'\in\mathcal{F}}\left|\mathcal{F}_{M\epsilon_{n'}}^{f'}\right|} \geq \frac{\exp\left(-B_2 n'\epsilon_{n'}^2\right)}{|\mathcal{F}| - \max_{f'\in\mathcal{F}}\left|\mathcal{F}_{M\epsilon_{n'}}^{f'}\right|}, \quad \forall n' > N \right\}. \tag{56}$$

In the remainder of the proof, we will suppose that $n > n^*$. For the given ground-truth $f \in \mathcal{F}$, if the event

$$\text{``} \mathcal{E}_{F|S_{\mathcal{X}}^n, S_{\mathcal{Y},f}^n}^\mu\left(f' : d_{\mathrm{TV}}^\mu(f', f) \leq M\epsilon_n \mid s_{\mathcal{X}}^n, s_{\mathcal{Y},f}^n\right) \geq \exp\left(-B_2 n\epsilon_n^2\right) \text{''} \tag{57}$$

holds true, then since $\dfrac{1 - \exp\left(-B_2 n\epsilon_n^2\right)}{\left|\mathcal{F}_{M\epsilon_n}^f\right|} \geq \dfrac{\exp\left(-B_2 n\epsilon_n^2\right)}{|\mathcal{F}| - \left|\mathcal{F}_{M\epsilon_n}^f\right|}$, we have that the disintegrated entropy

$H\left(F \mid s_{\mathcal{X}}^n, s_{\mathcal{Y},f}^n\right)$ is maximized when $\mathcal{E}_{F|S_{\mathcal{X}}^n, S_{\mathcal{Y},f}^n}^\mu\left(\cdot \mid s_{\mathcal{X}}^n, s_{\mathcal{Y},f}^n\right)$ is locally uniformly distribution on

$\mathcal{F}_{M\epsilon_n}^f$ and $\mathcal{F}\backslash\mathcal{F}_{M\epsilon_n}^f$ respectively. In concrete,

$$H\left(F \mid s_{\mathcal{X}}^n, s_{\mathcal{Y},f}^n\right)$$

$$\leq -\sum_{f'\in\mathcal{F}_{M\epsilon_n}^f}\frac{1-\exp\left(-B_2 n\epsilon_n^2\right)}{\left|\mathcal{F}_{M\epsilon_n}^f\right|}\log\frac{1-\exp\left(-B_2 n\epsilon_n^2\right)}{\left|\mathcal{F}_{M\epsilon_n}^f\right|} - \sum_{f'\in\mathcal{F}\backslash\mathcal{F}_{M\epsilon_n}^f}\frac{\exp\left(-B_2 n\epsilon_n^2\right)}{\left|\mathcal{F}\right|-\left|\mathcal{F}_{M\epsilon_n}^f\right|}\log\frac{\exp\left(-B_2 n\epsilon_n^2\right)}{\left|\mathcal{F}\right|-\left|\mathcal{F}_{M\epsilon_n}^f\right|}$$

$$\leq -\left(1-\exp\left(-B_2 n\epsilon_n^2\right)\right)\log\frac{1-\exp\left(-B_2 n\epsilon_n^2\right)}{\left|\mathcal{F}_{M\epsilon_n}^f\right|} - \exp\left(-B_2 n\epsilon_n^2\right)\log\frac{\exp\left(-B_2 n\epsilon_n^2\right)}{\left|\mathcal{F}\right|-\left|\mathcal{F}_{M\epsilon_n}^f\right|}$$

$$\leq -\left(1-\exp\left(-B_2 n\epsilon_n^2\right)\right)\log\frac{1-\exp\left(-B_2 n\epsilon_n^2\right)}{\max\limits_{f'\in\mathcal{F}}\left|\mathcal{F}_{M\epsilon_n}^{f'}\right|} - \exp\left(-B_2 n\epsilon_n^2\right)\log\frac{\exp\left(-B_2 n\epsilon_n^2\right)}{\left|\mathcal{F}\right|-\max\limits_{f'\in\mathcal{F}}\left|\mathcal{F}_{M\epsilon_n}^{f'}\right|}$$

$$= \left(1-\exp\left(-B_2 n\epsilon_n^2\right)\right)\log\max_{f'\in\mathcal{F}}\left|\mathcal{F}_{M\epsilon_n}^{f'}\right| + \exp\left(-B_2 n\epsilon_n^2\right)\log\left(\left|\mathcal{F}\right|-\max_{f'\in\mathcal{F}}\left|\mathcal{F}_{M\epsilon_n}^{f'}\right|\right) + \mathscr{H}_{\mathrm{b}}(\exp\left(-B_2 n\epsilon_n^2\right))$$

$$\leq \log\max_{f'\in\mathcal{F}}\left|\mathcal{F}_{M\epsilon_n}^{f'}\right| + \exp\left(-B_2 n\epsilon_n^2\right)\log\left|\mathcal{F}\right| + \exp\left(-\frac{B_2}{2}n\epsilon_n^2\right)$$

$$\leq \log\max_{f'\in\mathcal{F}}\left|\mathcal{F}_{M\epsilon_n}^{f'}\right| + \mathcal{O}\left(\exp\left(-\frac{B_2}{2}n\epsilon_n^2\right)\log\left|\mathcal{F}\right|\right).$$

$$(58)$$

On the other hand, if the event

$$\text{``}\mathcal{E}_{F\mid S_{\mathcal{X}}^n, S_{\mathcal{Y},f}^n}^{\mu}\left(f' : d_{\mathrm{TV}}^{\mu}(f',f)\leq M\epsilon_n \mid s_{\mathcal{X}}^n, s_{\mathcal{Y},f}^n\right)\geq\exp\left(-B_2 n\epsilon_n^2\right)\text{''} \qquad (59)$$

does not hold, then $H\left(F \mid s_{\mathcal{X}}^n, s_{\mathcal{Y},f}^n\right)$ is upper bounded by $\log|\mathcal{F}|$, then we have that the conditional entropy $H\left(F \mid S_{\mathcal{X}}^n, S_{\mathcal{Y}}^n\right)$ can be upper bounded as follows,

$$H\left(F \mid S_{\mathcal{X}}^n, S_{\mathcal{Y}}^n\right) = \mathop{\mathbb{E}}_{f\in\mathcal{E}_F^\mu} H\left(F \mid s_{\mathcal{X}}^n, s_{\mathcal{Y},f}^n\right)$$

$$\leq \mathop{\mathbb{E}}_{f\sim\mathcal{E}_F^\mu}\left[\log\max_{f'\in\mathcal{F}}\left|\mathcal{F}_{M\epsilon_n}^{f'}\right| + \mathcal{O}\left(\exp\left(-\frac{B_2}{2}n\epsilon_n^2\right)\log\left|\mathcal{F}\right|\right) + \exp\left(-B_1 n\epsilon_n^2\right)\log\left|\mathcal{F}\right|\right]$$

$$\leq \mathop{\mathbb{E}}_{f\sim\mathcal{E}_F^\mu}\left[\log\max_{f'\in\mathcal{F}}\left|\mathcal{F}_{M\epsilon_n}^{f'}\right| + \mathcal{O}\left(\exp\left(-B n\epsilon_n^2\right)\log\left|\mathcal{F}\right|\right)\right]$$

$$= \log\max_{f'\in\mathcal{F}}\left|\mathcal{F}_{M\epsilon_n}^{f'}\right| + \mathcal{O}\left(\exp\left(-B n\epsilon_n^2\right)\log\left|\mathcal{F}\right|\right),$$

$$(60)$$

where $B = \min\left\{\frac{B_2}{2}, B_1\right\}$. On the other hand, since $H\left(F \mid S_{\mathcal{X}}^n, S_{\mathcal{Y}}^n, X, Y\right)$ is lower bounded by $0$, we there by have the following upper bound on the conditional entropy difference,

$$H\left(F \mid S_{\mathcal{X}}^n, S_{\mathcal{Y}}^n\right) - H\left(F \mid S_{\mathcal{X}}^n, S_{\mathcal{Y}}, X, Y\right) \leq \log\max_{f'\in\mathcal{F}}\left|\mathcal{F}_{M\epsilon_n}^{f'}\right| + \mathcal{O}\left(\exp\left(-B n\epsilon_n^2\right)\log\left|\mathcal{F}\right|\right),$$

$$(61)$$

which completes the proof. $\qquad\square$

