# OpenReview forum: "On The Difficulty of Learning in Classification Problems: Optimality and Information-Theoretic Perspectives"
_ICLR.cc/2026/Conference — Submitted to ICLR 2026_

### Official Review · Reviewer_s5Lw · 2025-10-14

**Soundness:** 3
**Presentation:** 3
**Contribution:** 2
**Rating:** 2
**Confidence:** 4

**Summary:**

The paper investigates the intrinsic hardness of classification problems through an information-theoretic lens. The setting is defined by an input distribution and a prior over classifiers. The authors derive a closed-form optimal learner, the associated Bayes risk,
and new lower bounds on classification error expressed via the conditional mutual information $I(F;(X,Y)|S_n)$. The new bounds are claimed to be tighter and more practical than prior ones (Chen et al 2016), (Jeon & Roy 2022). Toy examples and asymptotic analyses support this claim, showing that previous bounds can be vacuous while theirs retain the correct convergence rates.

**Strengths:**

* By focusing on average-case risk rather than worst-case PAC or minimax risks, the work avoids the pessimism of classical bounds and connects more naturally to practical distributions.
* Concrete examples: Simple one-dimensional binary problems illustrate the bound’s behavior, highlighting its non-vacuity and the quadratic scaling gap relative to existing results.

**Weaknesses:**

Major:
* The presented results are somewhat incremental compared to the previous work of (Jeon & Roy 2022). Both works study the Bayesian posterior as the optimal learner, and use the mutual information $I(F;(X,Y)|S_n)$ to bound the expected classification risk. The insights provided by the two works are identical to a large extent. The main difference lies in the choice of loss function, where this paper uses a 0-1-like loss while (Jeon & Roy 2022) uses the KL divergence. However, this improvement may not be significant enough to be a substantial contribution.

Minor:
* One of the main theoretical results, Theorem 2, relies on some uncommon assumptions that lack proper justification (L435-440). It is not clear how they come and to what extent these conditions are met in practice.
* Some numerical studies on the examples provided could make the improvements more intuitive and strengthen the paper.

**Questions:**

* How would the results change if a different loss function is used, e.g., KL divergence or total variance between the two predictions? Currently, the bounds seem looser than (Jeon & Roy 2022) due to the extra $-1$ term or the square. Is it because of the choice of loss function? Will Theorem 1 recover the bound of (Jeon & Roy 2022) if the KL divergence is used?

---

> ### Author Response · Authors · 2025-11-22
>
> > The presented results are somewhat incremental compared to the previous work of (Jeon & Roy 2022). Both works study the Bayesian posterior as the optimal learner, and use the mutual information $I(F;(X,Y)\mid S_n)$ to bound the expected classification risk. The insights provided by the two works are identical to a large extent. The main difference lies in the choice of loss function, where this paper uses a 0-1-like loss while (Jeon & Roy 2022) uses the KL divergence. However, this improvement may not be significant enough to be a substantial contribution.
> >
> > How would the results change if a different loss function is used, e.g., KL divergence or total variance between the two predictions? Currently, the bounds seem looser than (Jeon & Roy 2022) due to the extra $-1$ term or the square. Is it because of the choice of loss function? Will Theorem 1 recover the bound of (Jeon & Roy 2022) if the KL divergence is used?
>
> We thank the reviewer for raising this comparison. We acknowledge that Jeon & Roy (2022) [1] is indeed related, but we would like to emphasize that their results do not downplay the significance of our contributions. While both works use an equivalent mutual information term at a high level, they apply it to fundamentally different problems, which lead to different mathematical results and distinct insights. In particular:
> 1. The choice of loss function changes the problem. Jeon & Roy (2022) analyze KL divergence between labeling functions, an unbounded divergence that aligns algebraically with mutual information. In contrast, we study 0–1 classification loss, which is bounded and directly reflects prediction error. These two risk notions have very different geometric and statistical behaviors, which lead to mathematically distinct analyses. Therefore, it is not meaningful to describe our result as “looser” since the analyses are simply not comparable.
> 2. The techniques are not interchangeable. Jeon & Roy obtain equalities because KL divergence decomposes additively and matches mutual information. Our bound is derived from Fano’s inequality. One cannot recover Jeon & Roy’s result by substituting KL for 0–1 loss in Theorem 1 since doing so would just change the problem itself.
> 3. The results themselves are genuinely different. Our theorem provides a lower bound on classification error, whereas Jeon & Roy’s result bounds expected KL divergence. These are inherently different objects, and neither dominates or generalizes the other. Thus, Theorem 1 is not intended and should not be expected to reproduce Jeon & Roy’s bound without altering the problem entirely.
> 4. Our work yields insight that do not appear in Jeon & Roy (2022) [1]. Most notably, our analysis uncovers an identifiability–agreement trade-off that jointly governs the hardness of classification problems. While Jeon & Roy also derive an equivalent conditional mutual information term, this structural interpretation is not discussed or developed in their work.
>
> [1] Hong Jun Jeon and Benjamin Van Roy. An information-theoretic framework for deep learning. In Alice H. Oh, Alekh Agarwal, Danielle Belgrave, and Kyunghyun Cho (eds.), Advances in Neural Information Processing Systems, 2022
>
> (Continued in the next comment.)

---

> ### Author Response · Authors · 2025-11-22
>
> > One of the main theoretical results, Theorem 2, relies on some uncommon assumptions that lack proper justification (L435-440). It is not clear how they come and to what extent these conditions are met in practice.
>
> Thank you for raising this concern. We provide here some intuitive insights into the regularity assumptions. As discussed in [2], which establishes the original posterior contraction theorem, the key idea is that the posterior can contract only if one can construct statistical tests that reliably distinguish the ground-truth labeling function $f$ from alternatives that lie at least a certain "distance" away (under an appropriate metric). The assumptions ensure that such tests exist. In particular:
> 1. $\log\mathcal{M}(\epsilon\_i,\mathcal{F}\_i,d^\mu_\text{TV})\leq i\epsilon_i^2$: The packing number is the maximum number of labeling functions that are $\epsilon_i$-far apart in the hypothesis class, measuring its inherent complexity. This inequality says that At distance scale $\epsilon_i$, the hypothesis class must not be too large, in other words, its "complexity" should be at most exponential of order $i\epsilon_i^2$. Without this, you cannot construct tests distinguishing $f_0$ from alternatives at distance $\epsilon_i$ with small type I/II error. If the hypothesis class has too many well-separated candidates, tests will fail, and the posterior cannot contract.
> 2. $\mathcal{E}^\mu_F(\mathcal{F}\backslash\mathcal{F}_i)\leq\exp(-i\epsilon_i^2(C+4))$: this inequality says that while the hypothesis class may still be not sufficiently "small", you can restrict attention to a “good” and "manageable" subspace $\mathcal{F}_i$ by assuring that the prior places most of its mass on this manageable subset. On the other hand, if the prior wastes too much probability mass outside $\mathcal{F}_i$, the posterior may "leak" there, slowing or even preventing its convergence.
> 3. $\min_{f\in\mathcal{F}\_\mu}\mathcal{E}^\mu_F(f)\geq\exp(-i\epsilon_i^2C)$: this is a special case of the following assumption in [2] under our "finite hypothesis class" condition: $\mathcal{E}^\mu_F(f':d^\mu_\text{KL}(f,f')\leq\epsilon_i^2)\geq\exp(-i\epsilon_i^2C)$. This requires that the prior places at least exponentially small mass on a KL neighborhood of the ground-truth. If no nearby distributions have enough prior mass, the likelihood cannot "pull" the posterior towards the ground-truth.
> 4. We do not currently have a deeper conceptual justification for the exact form of $(\epsilon_i)$, nor is one provided explicitly in [2]. Our understanding is that it is chosen largely for algebraic convenience in the proof. As noted in Lines 452–464, a carefully chosen form of $(\epsilon_i)$ ensures that the conditional entropy reduction term converges at approximately an exponential rate.
>
> We also appreciate your concern about whether these regularity assumptions can be met in practice. We agree that these assumptions are theoretical in nature and, like many conditions used in posterior-convergence theory, they may not hold exactly in realistic applications. They are imposed not because they perfectly describe real data, but because they identify the structural properties under which provable guarantees become possible.
>
> In other words, these assumptions serve as a “spherical cow” idealization. They isolate the essential statistical features (such as local complexity and prior concentration) that make posterior convergence provably achievable. This does not imply that real-world problems satisfy them exactly; rather, they help clarify why certain kinds of problems are learnable and what aspects of the hypothesis class or the prior distribution drive the contraction behavior. We agree that understanding how these assumptions manifest—or fail to manifest is an important direction for future work.
>
> [2] Subhashis Ghosal, Jayanta K. Ghosh, and Aad W. van der Vaart. Convergence rates of posterior distributions. The Annals of Statistics, 28(2):500 – 531, 2000.
>
> > Some numerical studies on the examples provided could make the improvements more intuitive and strengthen the paper.
>
> We appreciate your suggestion. We would like to note that in fact, for each provided example in our paper, we did perform numerical analyses and derived closed-form solutions for different metrics, including our bound and that of [3], which can be found in Section 4.4, Appendix A and Appendix B. This allowed us to compare the two bounds precisely and to conclude the sharpness of our bound. We hope this explanation will be helpful.
>
> [3] Xi Chen, Yuchen Zhang, et al. On bayes risk lower bounds. Journal of Machine Learning Research, 17(218):1–58, 2016.

---

> ### Comment · Reviewer_s5Lw · 2025-11-26
>
> Thank you for the detailed response. While I appreciate the authors' efforts to explain their core contributions, my main concern remains unresolved after the rebuttal. To me, this work is a revisit of (Jeon & Roy 2022) under a slightly different setting where the loss function is changed from KL divergence to 0-1 loss, with some new interpretations for the main MI quantities. I don't see substantial technical difficulties in extending (Jeon & Roy 2022) to this new setting. The identifiability–agreement trade-off may offer some novelty, but is not sufficient to be a substantial contribution. Therefore, I'm maintaining my initial score.

---

### Official Review · Reviewer_vkM9 · 2025-10-31

**Soundness:** 3
**Presentation:** 3
**Contribution:** 2
**Rating:** 4
**Confidence:** 2

**Summary:**

In this paper, the authors consider a new formulation of classification problems, specified by a fixed input distribution $\mu$ and a prior distribution $\mathcal{E}_{F}$ of hypotheses, to overcome the pessimism of the classic PAC learning model. The learner’s performance is then measured by the average classification risk rather than the worst-case risk. They identify the optimal learning algorithm and its (optimal) risk. Based on Fano's method, they provide new information-theoretic risk lower bound, which demonstrates advantage over previously-known lower bound on concrete examples. Finally, they also analyze the convergence of the lower bound (conditional entropy reduction quantity) for classes with finite cardinality.

**Strengths:**

This paper provides a new clear formulation of standard classification problem which is interesting and worth exploring. Under the new setting, the authors build a closed-form derivation of the optimal learner and its optimal risk, together with a new information-theoretic lower bound. One of this paper's shining points is that it reveals an innovative "identifiability–agreement" tradeoff characterization of the hardness of learning.

**Weaknesses:**

Minor comments:
(1): In Section 2 (Related works), since the preliminaries are provided later in Section 3, readers might get confused about, for example, what does a KL-divergence between two functions $f$ and $\hat{f}$ mean in Equation (3) (though they are clarified to be probability measures later in Section 3). In other words, I think it would be better to have Section 3 presented ahead of Section 2.

(2): In Section 5, the asymptotic result is not quite intuitively understandable, especially those additional assumptions and careful hyperparameter choice. (Please see questions below).

**Questions:**

1. The problem formulation is interesting if the authors can come up with a practical interpretation for it, namely, machine learning scenarios where the model is applicable (cases when we are interested in average risk). In that way, the new formulation becomes more meaningful and less artificial.

2. In learning theory, there is another line of works also focus on overcoming the pessimistic effect of the classic PAC model, called "universal learning" (see [1] and follow-up works). Did the authors compare them?

3. In Section 5, besides the finite cardinality (which is already quite limited), the result also requires some regularity assumptions. Can the authors explain how strong those assumptions are?

[1]: Bousquet, O., Hanneke, S., Moran, S., Van Handel, R., and Yehudayoff, A. (2021), “A theory of universal learning,” in Proceedings of the 53rd Annual ACM SIGACT Symposium on Theory of Computing, pp. 532–541.

---

> ### Author Response · Authors · 2025-11-22
>
> We thank you sincerely for your comments to our paper, and we truly appreciate the questions and concerns about our paper you have raised. Our response is as follows.
>
> > In Section 2 (Related works), since the preliminaries are provided later in Section 3, readers might get confused about, for example, what does a KL-divergence between two functions $f$ and $\hat{f}$ mean in Equation (3) (though they are clarified to be probability measures later in Section 3). In other words, I think it would be better to have Section 3 presented ahead of Section 2.
>
> Thank you for your suggestions on the presentation of our paper. We haved moved the introduction of notations from Section 3 to the end of Section 1, making the presentation clearer.
>
> > In Section 5, the asymptotic result is not quite intuitively understandable, especially those additional assumptions and careful hyperparameter choice. (Please see questions below).
> >
> > In Section 5, besides the finite cardinality (which is already quite limited), the result also requires some regularity assumptions. Can the authors explain how strong those assumptions are?
>
> Thank you for raising this concern. We provide here some intuitive insights into the regularity assumptions. As discussed in [1], which establishes the original posterior contraction theorem, the key idea is that the posterior can contract only if one can construct statistical tests that reliably distinguish the ground-truth labeling function $f$ from alternatives that lie at least a certain "distance" away (under an appropriate metric). The assumptions ensure that such tests exist. In particular:
> 1. $\log\mathcal{M}(\epsilon\_i,\mathcal{F}\_i,d^\mu_\text{TV})\leq i\epsilon_i^2$: The packing number is the maximum number of labeling functions that are $\epsilon_i$-far apart in the hypothesis class, measuring its inherent complexity. This inequality says that At distance scale $\epsilon_i$, the hypothesis class must not be too large, in other words, its "complexity" should be at most exponential of order $i\epsilon_i^2$. Without this, you cannot construct tests distinguishing $f_0$ from alternatives at distance $\epsilon_i$ with small type I/II error. If the hypothesis class has too many well-separated candidates, tests will fail, and the posterior cannot contract.
> 2. $\mathcal{E}^\mu_F(\mathcal{F}\backslash\mathcal{F}_i)\leq\exp(-i\epsilon_i^2(C+4))$: this inequality says that while the hypothesis class may still be not sufficiently "small", you can restrict attention to a “good” and "manageable" subspace $\mathcal{F}_i$ by assuring that the prior places most of its mass on this manageable subset. On the other hand, if the prior wastes too much probability mass outside $\mathcal{F}_i$, the posterior may "leak" there, slowing or even preventing its convergence.
> 3. $\min\_{f\in\mathcal{F}\_\mu}\mathcal{E}^\mu_F(f)\geq\exp(-i\epsilon_i^2C)$: this is a special case of the following assumption in [1] under our "finite hypothesis class" condition: $\mathcal{E}^\mu_F(f':d^\mu_\text{KL}(f,f')\leq\epsilon_i^2)\geq\exp(-i\epsilon_i^2C)$. This requires that the prior places at least exponentially small mass on a KL neighborhood of the ground-truth. If no nearby distributions have enough prior mass, the likelihood cannot "pull" the posterior towards the ground-truth.
> 4. We do not currently have a deeper conceptual justification for the exact form of $(\epsilon_i)$, nor is one provided explicitly in [1]. Our understanding is that it is chosen largely for algebraic convenience in the proof. As noted in Lines 452–464, a carefully chosen form of $(\epsilon_i)$ ensures that the conditional entropy reduction term converges at approximately an exponential rate.
>
> These assumptions are indeed strong, but they are standard regularity conditions required in posterior-convergence theory to make testing and convergence possible. They identify the structural properties under which guarantees can be proved. We believe that how to relax or empirically examine these assumptions is a valuable direction for future works.
>
> [1] Subhashis Ghosal, Jayanta K. Ghosh, and Aad W. van der Vaart. Convergence rates of posterior distributions. The Annals of Statistics, 28(2):500 – 531, 2000.
>
> (Continued in the next comment.)

---

> > ### Author Response · Authors · 2025-11-22
> >
> > > The problem formulation is interesting if the authors can come up with a practical interpretation for it, namely, machine learning scenarios where the model is applicable (cases when we are interested in average risk). In that way, the new formulation becomes more meaningful and less artificial.
> >
> > There are many practical scenarios using the average risk over the models. For example, consider a simple movie rating task, where
> > * each movie is an input sample $X\sim \mu$,
> > * each film critic is represented by a classifier $F\sim \mathcal{E}_F$, whose output domain is a finite set $\mathcal{Y}:=\{1,2,3,4,5\}$.
> > * A critic $F$ is randomly invited, and a set $S_{\mathcal{X}^n}$ of $n$ movies is randomly drawn. The critic $F$ then gives ratings/labels  $S_{\mathcal{Y}^n}$ for those movies. The goal of this task is to learn a rating function (or a classifier) $\hat{F}$.
> >
> > In this setting, it is more reasonable to define the risk as the average error between $\hat{F}$ and all film critics, rather than the error between $\hat{F}$ and the critic with the most random ratings (representing the worst-case scenario).
> >
> > > In learning theory, there is another line of works also focus on overcoming the pessimistic effect of the classic PAC model, called "universal learning" (see [2] and follow-up works). Did the authors compare them?
> >
> > We thank the reviewer for pointing out this related line of work. We now include it in the introduction section:
> > Universal learning is a novel framework that studies the asymptotic convergence behavior of the error upper bounds for a specific data distribution. The work of [2] shares a similar motivation as the present paper. It argues that the classic PAC learning (or called uniform learning in [2]) is overly pessimistic due to its reliance on the worst-case data distribution. However, our approach and perspective differ from [2] in several respects:
> > * The theoretical results of [2] depend on a given data distribution and a given hypothesis class. Our results are average-case, depending on a given data distribution and a prior on a class of hypothesis.
> > * The problems studied in [2] and in our work are fundamentally different: [2] studies the asymptotic convergence behavior of the risk upper bounds. We study the risk lower bound.
> >
> > [2]: Bousquet, O., Hanneke, S., Moran, S., Van Handel, R., and Yehudayoff, A. (2021), “A theory of universal learning,” in Proceedings of the 53rd Annual ACM SIGACT Symposium on Theory of Computing, pp. 532–541.

---

### Official Review · Reviewer_vCPZ · 2025-11-01

**Soundness:** 3
**Presentation:** 4
**Contribution:** 3
**Rating:** 8
**Confidence:** 3

**Summary:**

This paper proposes a new general information-theoretic lower bound for classification problems. The bound is different from the “worst-case” sense of lower bounds in the PAC framework, which typically comes up with a single ground-truth distribution for which learning is hard. Instead, the regime is on “average” over a distribution $\varepsilon_F$ of ground-truth hypothesis.

The proposed lower bound is in a similar framework to Chen et al. 2016, where the risk of any learning algorithm is measured with respect to a randomly sampled dataset of $n$ points, a randomly sampled ground truth hypothesis $f \sim \varepsilon_F$, and a random hypothesis sampled from the learner’s output distribution trained on the input dataset. The paper first shows that even in very simple scenarios with, for example, threshold classifiers, the lower bound of Chen et al. can become vacuous. The paper argues that this is because while the bound from Chen et al. considers the identifiability of hypotheses (captured by the conditional entropy of the hypothesis set and sampled dataset), the same bound does _not_ consider the agreement of potential hypotheses on a new sampled datapoint.

The paper then proposes that the agreement (and identifiability) can indeed be captured by considering the conditional entropy of the hypothesis set and sampled dataset in addition to the additional knowledge gained by sampling one further datapoint. This simplifies to eq (22) in Theorem 1, which cleanly captures these two notions.

Finally, in Theorem 2, the paper studies what happens as n-> inf. In learning theory, we often assume that everything is learnable (or at least, identifiable) with an infinite number of samples, so one would hope that this behavior is captured in the lower bound. Indeed, this is captured by how quickly the lower bound diminishes as n increases, which the theorem characterizes.

**Strengths:**

Note that I did not read the appendix, and do not work in the “information theory for lower bounds in learning” area. Although theoretical, the paper is written in a very accessible manner, which makes understanding the intuition behind the results very simple. In fact, the way that the paper is presented makes the result seem so clear in hindsight that it is surprising it has not been done before. This is a strength, and I take it to mean that the paper is natural and important.

I also like the decomposition in the lower bound between the “agreement” of hypotheses and their “identifiability”. The motivation in the first threshold / interval classification problem and the failure of prior work (Chen et al.) to capture this is very clear.

**Weaknesses:**

The examples given in the main paper show how the proposed bound is an improvement over prior work. However, can one come up with settings where the proposed bound is loose? What do these settings look like? This may help show further limitations of the lower bound.

An additional weakness is that some form of proof sketch should probably be given for Theorem 1, since it is the main result of the paper. How does the proof go, or what are the intuitions behind some of the steps? This may be especially important for readers like me who are unfamiliar with these kinds of information-theoretic arguments.

There may also be the question of whether having only Theorem 1 + 2 is “enough” for the paper to be accepted. I believe that the story is very nice and intuitive, and I don’t see this as a weakness.

**Questions:**

1. This is potentially somewhat tangential. Are the conditional entropy decompositions related in any way to the aleatoric / epistemic decompositions from the work of Ahdritz et al. 2025 (who also use conditional entropy)? Can I interpret the lower bound as saying how quickly we can decrease the “epistemic uncertainty” of the model with each additional data point? Or, perhaps interpreting eq (22) directly, part (1) reflects the “epistemic” (reducible) uncertainty, and part (2) reflects the “aleatoric” (irreducible) uncertainty in nature itself. Thus, learning can be hard either because we do not have enough samples, or because nature itself is random.
2. Is the proposed decomposition related to any work in active learning? There, the goal is often to reduce the uncertainty a maximal amount for each added datapoint to the dataset. For example, Castro and Nowak (Minimax Bounds for Active Learning, 2007?).

Ahdritz et al. Provable Uncertainty Decomposition via Higher-Order Calibration. ICLR 2025.

Suggestions
1. Line 66: “Uses KL divergence as a measure of risk , making the resulting lower bound largely irrelevant for practical considerations.” Maybe one more sentence on why this is the case? As someone who does not work on lower bounds, it is not immediately clear what this means. Coming back, I found that line 222-227 explains this more somewhat. Perhaps turn 222-227 into a remark which you can reference in line 66.
2. For those who do not work in lower bounds or information theory, perhaps the use of Fano’s inequality and how it differs from applications in prior work could be helpful.

Typos
1. Line 219: “unstrained” -> “unconstrained”?

---

> ### Author Response · Authors · 2025-11-22
>
> We thank you sincerely for your comments to our paper, and we truly appreciate the questions and concerns about our paper you have raised. Our response is as follows.
>
> > The examples given in the main paper show how the proposed bound is an improvement over prior work. However, can one come up with settings where the proposed bound is loose? What do these settings look like? This may help show further limitations of the lower bound.
>
> Thank you for this important question regarding the sharpness of our bound. Our lower bound is derived directly from the standard Fano's inequality, which is well known to give sharp lower bounds in multi-hypothesis settings [1]. In contrast, several existing works rely on generalized variants of Fano's inequality or on alternative tools such as data processing relaxations that typically induce additional slack terms. Thus, our analysis conceptually inherits the sharpness of the standard Fano's inequality.
>
> That said, we acknowledge that no single inequality is universally tight in all problem configurations. Although so far we have not yet identified a concrete example where our bound is provably looser than others, we expect such cases may arise in highly structured or specialized settings. Characterizing these regimes is, in our view, an interesting but challenging direction for future works and lies outside the scope of the present paper.
>
> [1] Cover, T. M., & Thomas, J. A. (2006). Elements of information theory (2nd ed.) Sec. 2.10. Wiley.
>
> > An additional weakness is that some form of proof sketch should probably be given for Theorem 1, since it is the main result of the paper. How does the proof go, or what are the intuitions behind some of the steps? This may be especially important for readers like me who are unfamiliar with these kinds of information-theoretic arguments.
>
> Thank you for the valuable suggestion! We have added a brief sketch of proof of Theorem 1 in Section 4.3 of the revised version.
>
> > This is potentially somewhat tangential. Are the conditional entropy decompositions related in any way to the aleatoric / epistemic decompositions from the work of Ahdritz et al. 2025 (who also use conditional entropy)? Can I interpret the lower bound as saying how quickly we can decrease the “epistemic uncertainty” of the model with each additional data point? Or, perhaps interpreting eq (22) directly, part (1) reflects the “epistemic” (reducible) uncertainty, and part (2) reflects the “aleatoric” (irreducible) uncertainty in nature itself. Thus, learning can be hard either because we do not have enough samples, or because nature itself is random.
>
> We appreciate your thoughtful observation regarding this connection. After reading [2], we agree that there is a meaningful conceptual link between our decomposition in Eq. (22) and their treatment of uncertainty. In particular, the second term $H(Y\mid F,X)$ in Eq. (22) captures the intrinsic randomness in nature’s labeling process and can reasonably be viewed as representing aleatoric uncertainty.
>
> However, we would like to clarify that the first term in our decomposition in Eq. (22) might not be directly interpreted as epistemic uncertainty. In our framework, this term is entirely determined by the statistical relationship between the observed data and the ground-truth labeling function drawn by nature, and it is algorithm-independent. On the other hand, epistemic uncertainty depends not only on the problem structure but also on the learning algorithm used. Thus, although our decomposition resembles the aleatoric–epistemic split, only the second term aligns with aleatoric uncertainty, while the first term reflects a similar but different and purely information-theoretic notion.
>
> [2] Ahdritz et al. Provable Uncertainty Decomposition via Higher-Order Calibration. ICLR 2025.
>
> (Continued in the next comment.)

---

> ### Author Response · Authors · 2025-11-22
>
> > Is the proposed decomposition related to any work in active learning? There, the goal is often to reduce the uncertainty a maximal amount for each added datapoint to the dataset. For example, Castro and Nowak (Minimax Bounds for Active Learning, 2007?).
> Yes, the proposed Theorem 1 provides theoretical guidance for data selection in active learning.
> Consider the standard pool-based active learning problem, assume that the initial dataset contains $n$ data pairs, denoted by $s\_n:=\{(x_i,y_i)\}_{i=1}^n$. At the $t$-th step, the learner selects a new point $(x\_{n+t},y\_{n+t})$ (labeled by an oracle) and updates the dataset to $s\_{n+t}:=\{s\_{n+t-1}, (x\_{n+t},y\_{n+t})\}$.
>
> The proposed Theorem 1 can also be expressed in the following form: $R\geq \mathbb{E}\_{s\_n}[H(F\mid (S_{\mathcal{X}}^n,S_{\mathcal{Y}}^n)=s_n)-H(F\mid X,Y, (S_{\mathcal{X}}^n,S_{\mathcal{Y}}^n)=s_n)]+H(Y\mid F,X)$ $:= \Lambda_n(s_n)$,
> where the similar proof can be found in [3]. In the active learning setting, at step $t$, the proposed Theorem 1 suggests that the new data point is selected by solving $\min_{x_{n+t},y_{n+t}} \Lambda_{n+t}(s_{n+t-1} \cup (x_{n+t},y_{n+t}))$. Since such a selection yields the smallest achievable risk lower bound among all learning algorithms.
>
> [3] Z Dong, Z Liu, Y Mao. On the Hardness of Unsupervised Domain Adaptation: Optimal Learners and Information-Theoretic Perspective. CoLLAs 2025.
>
> (Continued in the next comment.)

---

> ### Author Response · Authors · 2025-11-22
>
> > Line 66: “Uses KL divergence as a measure of risk , making the resulting lower bound largely irrelevant for practical considerations.” Maybe one more sentence on why this is the case? As someone who does not work on lower bounds, it is not immediately clear what this means. Coming back, I found that line 222-227 explains this more somewhat. Perhaps turn 222-227 into a remark which you can reference in line 66.
>
> Thank you for pointing out this presentational flaw. We have added a reference in line 67 (in the revised version) to our explanation of the irrelevancy between KL divergence and 0-1 loss in classification problems.
>
> > For those who do not work in lower bounds or information theory, perhaps the use of Fano’s inequality and how it differs from applications in prior work could be helpful.
>
> Thank you for the suggestion. We have added the connection between Fano’s inequality and our lower bound and how it differs from prior work in Appendix E.
>
> Our contribution is to make this Bayesian perspective concrete specifically for classification problems, by modeling labeling functions as random parameters that generates labels for input data drawn from an input distribution $\mu$. Prior works such as [4] do not formulate a model as concrete as ours on classification problems and therefore do not obtain our clean, explicit interpretation. Our concrete formulation to align classification problems with Fano's inequality is also what enables our insight into the identifiability–agreement trade-off.
>
> [4] Xi Chen, Yuchen Zhang, et al. On bayes risk lower bounds. Journal of Machine Learning Research,
> 17(218):1–58, 2016.
>
> > Line 219: “unstrained” -> “unconstrained”?
>
> Thank you for catching this typo, we have corrected it.

---

### Official Review · Reviewer_2Esk · 2025-11-02

**Soundness:** 3
**Presentation:** 2
**Contribution:** 2
**Rating:** 4
**Confidence:** 4

**Summary:**

This paper studies Bayesian classification with a fixed input distribution mu and a known prior E_F over label-generating classifiers F. The goal is to lower bound the Bayes risk, i.e., the best achievable error after observing n training examples. The main result decomposes the residual uncertainty about a test label Y (given the training set S and test input X) into two parts: I(F;X,Y|S), which is the one-example information gain about the ground-truth classifier that remains after training, and H(Y|F,X), which captures intrinsic label noise. A Fano-type argument turns this residual uncertainty into a lower bound on the misclassification error. As n grows and F becomes identified, I(F;X,Y|S) shrinks and the bound becomes dominated by H(Y|F,X).

**Strengths:**

The decomposition is clear and interpretable: it separates task ambiguity (what we still do not know about F after seeing S) from irreducible noise (randomness in Y given F and X). The framing is consistent with Bayesian learning, and the result reads naturally.

**Weaknesses:**

Technical novelty appears limited. The core derivation relies on standard conditional independence, chain rules, and a textbook Fano inequality; the Appendix E proof is essentially careful bookkeeping.  Lemma 1 and Proposition 1 follow almost immediately from the definition of Bayes risk. Lemma 2 is a standard Fano inequality. please position what is genuinely new relative to existing lower-bound techniques.

The practical motivation is lacking: the paper suggests this “sidesteps pessimistic PAC bounds,” but it is not yet clear when the proposed Bayes-risk lower bound yields strictly sharper or more actionable insights. The knowledge assumptions are also strong: do we actually know mu and E_F in practice, and if not, how informative is the bound under prior or distributional misspecification?

Minor comments

- In figure 2 what is F and hat F?

- The citation is wrong for lower bound on VC classes. The lower bound comes from this work:
A. Ehrenfeucht, D. Haussler, M. Kearns, and L. G. Valiant. A general lower bound on the number of examples needed for learning.

**Questions:**

see above

---

> ### Author Response · Authors · 2025-11-22
>
> We thank you sincerely for your comments to our paper, and we truly appreciate the questions and concerns about our paper you have raised. Our response is as follows.
>
> > Technical novelty appears limited. The core derivation relies on standard conditional independence, chain rules, and a textbook Fano inequality; the Appendix E proof is essentially careful bookkeeping. Lemma 1 and Proposition 1 follow almost immediately from the definition of Bayes risk. Lemma 2 is a standard Fano inequality. please position what is genuinely new relative to existing lower-bound techniques.
>
>
> We agree that the mathematical tools we use such as mutual-information chain rules, conditional independence in Markov chains, and the Fano's inequality, etc. are standard. Our contributions do not lie in introducing new mathematical tools, but in applying these existing tools to analyzing classification problems so as to bring out fundamental discovery and new insights. Specifically our contributions include
>
> 1. $\textbf{Novel problem formulation.}$ We model a classification problem as parameter inference over a random ground-truth labeling function, rather than as learning a fixed hypothesis. This perspective creates clear conceptual links between classification problems and classical parameter-estimation theory in statistics and information theory.
> 2. $\textbf{Average-case risk analysis.}$ Our lower bound is derived based on an average notion of risk induced by nature’s prior over hypotheses. This avoids the pessimism of worst-case PAC-style bounds. In our response to the next comment, we present an example illustrating how this perspective yields more realistic characterizations of classification hardness.
> 3. $\textbf{Identifiability–agreement trade-off.}$ Our analysis uncovers a tension between structural concepts in classification: how distinguishable the labeling functions in the  hypothesis class are with respect to each other (identifiability), and to what extent they agree on the data distribution (agreement). To our knowledge, this tension is for the first time presented and it offers insighful explanations as to why some classification problems are intrinsically harder than others.
>
> Finally, we respectfully note that exploiting simple tools in research should not be taken as a weakness or penalized. Albert Einstein once advised, "everything should be made as simple as possible, but not simpler.” Important results explained using simple languages allows them to be accessed by much wider communities and in fact should be taken as a strength.
>
> (Continued in the next comment.)

---

> ### Author Response · Authors · 2025-11-22
>
> > The practical motivation is lacking: the paper suggests this “sidesteps pessimistic PAC bounds,” but it is not yet clear when the proposed Bayes-risk lower bound yields strictly sharper or more actionable insights.
>
> Thank you for raising this concern. Through the following example, we illustrate when the Bayes-risk perspective is more informative than the worst-case perspective.
>
> Let $\mathcal{X}=\{x_1,\cdots,x_M\}$ and $\mathcal{Y}=\{0,1\}$. Let $\mathcal{F}=\{f_1,\cdots,f_M\}$ be a class of hard classifiers where each $f_i$ predicts label $1$ at only point $x_i$ and label $0$ otherwise. Suppose that the input distribution $\mu$ is such that $\mu(x)=\epsilon$ if $x=x_M$ and $\mu(x)=\frac{1-\epsilon}{M-1}$ if $x\neq x_M$. In particular, $\epsilon$ is a positive number smaller than $1/M$. Let $\mathcal{E}_F$ be uniform on $\mathcal{F}$.
>
> To correctly identify any selected ground-truth $f_i$, the learner must observe $x_i$ at least once. Otherwise, all labels appear as $0$, and the Bayes optimal classifier predicts $0$ everywhere. A direct calculation gives:
> \begin{equation}
> R_\text{worst-case}=\max\{\beta(1-\beta)^n,\epsilon(1-\epsilon)^n\},
> \end{equation}
> where $\beta=\frac{1-\epsilon}{M-1}$. On the other hand, the Bayes risk is:
> \begin{equation}
> R_\text{Bayes}=\dfrac{M-1}{M}\beta(1-\beta)^n+\dfrac{1}{M}\epsilon(1-\epsilon)^n.
> \end{equation}
>
> It is easy to see that $R_\text{Bayes}$ is smaller than $R_\text{worst-case}$, since $R_\text{Bayes}$ is the weighted sum of $\beta(1-\beta)^n$ and $\epsilon(1-\epsilon)^n$, while $R_\text{worst-case}$ is the max of the two.
>
> It is also possible to compare $R_\text{Bayes}$ with $R_\text{worst-case}$, more quantitatively. To that end, denote
> \begin{equation}
> \gamma: = \dfrac{1-\dfrac{1-\epsilon}{M-1}}{1-\epsilon}
> \end{equation}
> It is easy to verify that $\gamma <1$ (due to $\epsilon <1/M$). Let $N^\*$ be the smallest value $n$ for which $\dfrac{\beta}{\epsilon}\cdot \gamma^n <1$. Then the following can be shown without great difficulty: for all $n>N^*$,
> \begin{equation}
> R_\text{Bayes} < R_\text{worst-case} \cdot(\dfrac{1}{M}+ \dfrac{\beta}{\epsilon}\cdot \gamma^n)
> \end{equation}
>
> Note that $\dfrac{\beta}{\epsilon} \gamma^n$ vanishes exponentially with sample size $n$, it is clear that for sufficiently large $n$, $R_\text{Bayes}$ is at most $\dfrac{1}{M}R_\text{worst-case}$. That is, $R_\text{Bayes}$ is significantly lower than $R_\text{worst-case}$, even for a moderate size $M$ of the input space.
>
> (Continued in the next comment.)

---

> ### Author Response · Authors · 2025-11-22
>
> > The knowledge assumptions are also strong: do we actually know mu and E_F in practice, and if not, how informative is the bound under prior or distributional misspecification?
>
> Thank you for raising this point; we agree that it highlights an important aspect of the framework, and while we have yet to derive a mathematically strict conclusion or solution of this question, below we are happy to provide some intuitive insights.
>
> From a learning-theoretic perspective, the assumption that the learner knows the true prior over hypotheses is not particularly strong. The notion of a prior over the hypothesis space parallels the notion of concept class (or hypothesis class) in the PAC-learning framework: such choices of formulation aim at model the uncertainty of the learner about the ground-truth classifier. We do agree that assuming access to the exact data distribution $\mu$ is an unusually strong assumption, but having this assumption merely serves to make the derivations cleaner. Extending the framework to one that also accounts for uncertainty in $\mu$ (by placing a prior on $\mu$), although involving certain technicality, entails little additional difficulties.
>
> From a practical standpoint, we acknowledge that neither the true prior $\mathcal{E}_F$ nor the true data distribution $\mu$ is known. Note that $(\mu,\mathcal{E}_F)$ jointly define a distribution over $\mathcal{X}\times\mathcal{F}$, and because each classifier induces a conditional distribution over $\mathcal{Y}$, the same pair also implicitly defines a distribution over the data space $(\mathcal{X}\times\mathcal{Y})$. Since our results concern average-case risk, this change of point of view does not change the nature of our analysis.
>
> We may first analyze this problem from the perspective of domain adaptation. This perspective naturally connects to domain adaptation, where a model trained on a source data distribution is evaluated on a target data distribution. Prior work has studied in depth how discrepancies between the two distributions affect performance [1][2][3]. In our setting, we may regard $(\mu,\mathcal{E}_F)$ as the source distribution and any approximation $(\mu',\mathcal{E}_F')$ of this pair as the target distribution. Existing domain adaptation techniques can then be applied to analyze how such distributional misspecification impacts the validity of our bounds.
>
> (Continued in the next comment.)

---

> ### Author Response · Authors · 2025-11-22
>
> On the other hand, we may also interprete this problem from a geometric point of view. To simplify the presentation, let's suppose that $\mathcal{H}$ is the set of all hard classifiers of a classification setting. Let the true prior $\mathcal{E}\_F$  be a uniform distribution on a subset $\mathcal{F}$ of $\mathcal{H}$. Let $\mathcal{E}\_F'$ be our guessed prior which is uniform on another subset $\mathcal{F}'$ of $\mathcal{H}$. After observing a training sample $S$, the posteriors $\mathcal{E}\_{F\mid S}$ and $\mathcal{E}\_{F\mid S}'$ with respect to the two priors will be uniform distributions supported on $\mathcal{F}\_S\subseteq\mathcal{F}$ and $\mathcal{F}\_S'\subseteq\mathcal{F}'$ respectively, where $\mathcal{F}\_S\subseteq\mathcal{F}$ and $\mathcal{F}\_S'\subseteq\mathcal{F}'$ denotes the subsets of classifiers that have consistent predictions on examples in sample $S$. The optimal predictor will be the aggregated classifier $\bar{f}\_{\mathcal{E}\_{F\mid S}}$ and $\bar{f}\_{\mathcal{E}\_{F\mid S}'}$, and the Bayes risks with respect to true prior and guessed prior naturally follow. We are interested with how to characterize $R_\text{Bayes}(\bar{f}\_{\mathcal{E}\_{F\mid S}})-R_\text{Bayes}(\bar{f}\_{\mathcal{E}\_{F\mid S}'})$.
>
> The core idea in this point of view is that we may analyze the following three regions: $\mathcal{F}\_S\backslash\mathcal{F}\_S'$, $\mathcal{F}\_S'\backslash\mathcal{F}\_S$ and $\mathcal{F}\_S\cap\mathcal{F}\_S'$. We can suppose a following case: suppose that the classifiers in $\mathcal{F}\_S\backslash\mathcal{F}\_S'$ have the best performance with respect to the true posterior $\mathcal{E}\_{F\mid S}$, suppose that the classifiers in $\mathcal{F}\_S'\backslash\mathcal{F}\_S$ have the worst performance with respect to the guessed posterior $\mathcal{E}\_{F\mid S}'$. The classifiers in $\mathcal{F}\_S\cap\mathcal{F}\_S'$ need not be carefully characterized since there effect on $R_\text{Bayes}(\bar{f}\_{\mathcal{E}\_{F\mid S}})-R_\text{Bayes}(\bar{f}\_{\mathcal{E}\_{F\mid S}'})$ will be cancelled (due to our assumption that both the true prior and the guessed prior are uniform distributions). The core element remains in this analyze is to study how the descrepancy between $\mathcal{E}\_F$ and $\mathcal{E}\_F'$ with respect to certain metrics can control the size of the regions $\mathcal{F}\_S\backslash\mathcal{F}\_S'$, $\mathcal{F}\_S'\backslash\mathcal{F}\_S$ and $\mathcal{F}\_S\cap\mathcal{F}\_S'$. Also, it would be important to extend the corresponding results to more general scenarios (with more general assumptions on the forms of $\mathcal{E}\_F$ and $\mathcal{E}\_F'$).
>
> We view a formal treatment of these questions as an interesting and challenging direction for future work. We again greatly thank you for raising this concern, as it touches on probably one of the most important yet not sufficiently studied issues for theoretical research in this area.
>
> [1] Ben-David, S. and Urner, R., 2012, October. On the hardness of domain adaptation and the utility of unlabeled target samples. In International Conference on Algorithmic Learning Theory (pp. 139-153). Berlin, Heidelberg: Springer Berlin Heidelberg.
>
> [2] Dong, Z., Liu, Z. and Mao, Y., 2025. On the Hardness of Unsupervised Domain Adaptation: Optimal Learners and Information-Theoretic Perspective. arXiv preprint arXiv:2507.06552.
>
> [3] Hanneke, S. and Kpotufe, S., 2019. On the value of target data in transfer learning. Advances in Neural Information Processing Systems, 32.
>
> (Continued in the next comment.)

---

> ### Author Response · Authors · 2025-11-22
>
> > In figure 2 what is F and hat F?
>
> In figure 2, $F$ denotes the random ground-truth labeling function drawn by nature according to $\mathcal{E}_F$, and $\hat{F}$ denotes the random predictor sampled from the learned distribution $\mathscr{A}(\cdot\mid S_n)$ over the hypothesis class $\mathcal{F}$ during evaluation. We use capital letters to emphasize that these objects are random with respect to their corresponding distribution. Also, in lines 180 and 186, the lowercase versions refer to their realizations. A summary of notation is provided between lines 112–117. We have also added a clarifying remark in the caption of figure 2.
>
> > The citation is wrong for lower bound on VC classes. The lower bound comes from this work: A. Ehrenfeucht, D. Haussler, M. Kearns, and L. G. Valiant. A general lower bound on the number of examples needed for learning.
>
> Thank you for pointing this out! We appreciate your correction. We have updated the manuscript: the discussion of PAC lower bounds in line 36 now also cites [4] as the source of the VC-class lower bound. We have retained the original reference [5] as the citation of the background of the PAC Learning framework in general.
>
> [4] Ehrenfeucht, A., Haussler, D., Kearns, M. and Valiant, L., 1989. A general lower bound on the number of examples needed for learning. Information and Computation, 82(3), pp.247-261.
>
> [5] Leslie G Valiant. A Theory of the Learnable. Communications of the ACM, 27(11):1134–1142, 1984.

---

### Author Response · Authors · 2025-12-02
**General Response**

We would like to thank the ICLR organization committee for your great effort promptly handling the data leakage issue. We would like to use the following public comments to summarize our responses and explanations to several major concerns raised during the review process. We hope by doing so we can make our rebuttals easier to follow.

We begin by highlighting several strengths of our work that were noted by the reviewers:
* Clear and accessible presentation that makes the decomposition intuitive.
* The decomposition cleanly separates task ambiguity from irreducible noise.
* Natural Bayesian framing and a new, clean formulation of classification problems.
* Novel identifiability–agreement trade-off capturing learning hardness.
* Average-case risk perspective avoids pessimism of PAC/minimax bounds.
* Concrete examples demonstrate sharpness and improve over prior work.

We then summarize our responses to the major concerns.

1. Technical novelty seems limited because the derivations rely on standard tools, and it is unclear what new contributions this work provides beyond existing lower-bound techniques. (2Esk)

For this concern, our response is that although we use standard tools, our contributions lie in how these tools are applied to classifications. We introduce a novel formulation where the ground-truth labeling function is random, enabling conceptual connections to parameter estimation. We derive average-case lower bounds, which avoid the pessimism inherent to worst-case PAC analyses, and we uncover a new identifiability–agreement trade-off that explains intrinsic sources of problem difficulty. Finally, we emphasize that using simple, well-established tools is a strength, not a weakness, because it leads to clearer and more broadly accessible theoretical insights.

2. Why should we care about reducing the pessimism of worst-case risk lower bounds (PAC learning bounds, minimax bounds)? Has our work addressed this issue, and how? (2Esk and vkM9)

In our response, we have provided a realistic example to show that in many real settings—such as learning a movie-rating function from randomly selected critics—it is more natural to measure risk as the average error against typical critics rather than the error against the single most erratic critic, illustrating why average-case risk is often the meaningful quantity. Details of this example can be found in our second comment to reviewer vkM9. On the other hand, we have provided an example to show that in a simple classification setup where each hypothesis labels exactly one point in a finite input alphabet as positive, the Bayes risk can be dramatically smaller than the worst-case risk. Because identifying the true hypothesis requires observing a specific rare input, the worst-case risk is dominated by the hypothesis associated with the least likely point, while the Bayes risk averages over all hypotheses under a uniform prior. A direct calculation shows that for sufficiently large sample size $n$, the Bayes risk becomes at most $\frac{1}{M}$ (where $M$ is the size of the input alphabet) of the worst-case risk, demonstrating that the average-case perspective yields a far less pessimistic and more realistic assessment of the problem’s difficulty. Details of the second example can be found in our second comment to reviewer 2Esk.

3. The assumptions requiring knowledge of the true data distribution $\mu$ and the prior $\mathcal{E}_F$ over the hypothesis class may be unrealistic in practice, raising questions about how meaningful the bound remains when these quantities are misspecified. (2Esk)

Our response is that although knowing the exact prior $\mathcal{E}_F$ and data distribution $\mu$ is a strong assumption, this is standard in theoretical learning frameworks, and the model can naturally be extended to include uncertainty over $\mu$ as well. We have outlined a geometric viewpoint showing how differences between the true and guessed priors influence the Bayes risk through changes in the posterior-supported classifier sets, suggesting a principled way to study misspecification potentially. A full mathematical treatment of these questions remains an interesting and challenging direction for future work.

4. Do there exist settings where the proposed bound becomes loose, and what such scenarios would look like, as identifying them could reveal further limitations of the method. (vCPZ)

Our response is that our bound inherits the sharpness of the standard Fano’s inequality from which it is derived—unlike many prior approaches that introduce slack through generalized variants—but while we have not yet found concrete settings where it becomes loose, such cases may exist in highly specialized problem structures and represent an interesting direction for future work.

(Continued in the next comment.)

---

> ### Author Response · Authors · 2025-12-02
>
> 5. Does the conditional-entropy decomposition align with aleatoric and epistemic uncertainty as in Ahdritz et al. (2025)? Do parts of our bound can be interpreted respectively as reducible (epistemic) and irreducible (aleatoric) sources of difficulty in learning? (vCPZ)
>
> We agree that our decomposition has a conceptual connection to aleatoric–epistemic uncertainty, and in particular the second term $ H(Y\mid F,X)$ naturally corresponds to aleatoric uncertainty arising from nature’s inherent randomness. However, the first term in our decomposition is not directly epistemic uncertainty, as it is entirely determined by the statistical relationship between data and the ground-truth labeling function rather than by properties of a learning algorithm, making it an information-theoretic quantity that only partially resembles epistemic uncertainty.
>
> 6. Does our decomposition connect to active learning, where each selected data point aims to maximally reduce uncertainty, as in works such as Castro and Nowak (2007)? (vCPZ)
>
> Our response is that Theorem 1 naturally guides active learning because it characterizes how the expected risk lower bound changes with each newly added labeled data point. In particular, at each step the learner should select the next point that minimizes the updated lower bound, thereby achieving the greatest theoretical reduction in uncertainty.
>
> 7. The asymptotic result in Theorem 2 Section 5 depends on restrictive and insufficiently justified regularity assumptions, making it unclear how these conditions arise or how often they hold in practice. (vkM9 and s5Lw)
>
> Our response is that the regularity assumptions arise directly from the classical posterior contraction framework of [1], where they ensure the existence of statistical tests that can reliably distinguish the true labeling function from alternatives of sufficient distance. Each assumption controls either the local complexity of the hypothesis class or the concentration of the prior, all of which are standard idealizations used to identify the structural conditions under which provable contraction is possible. While these assumptions may not hold exactly in practice, they clarify the statistical features that make learning feasible, and understanding their practical manifestations remains an important direction for future work.
>
> [1] Subhashis Ghosal, Jayanta K. Ghosh, and Aad W. van der Vaart. Convergence rates of posterior distributions. The Annals of Statistics, 28(2):500 – 531, 2000.
>
> 8. Have we compared our approach to the “universal learning’’ line of work that also seeks to address the pessimism of the classical PAC model. (vkM9)
>
> Our response is that while universal learning shares the motivation of avoiding worst-case pessimism, it differs fundamentally from our work because it analyzes asymptotic upper bounds for a fixed data distribution and hypothesis class, whereas we study average-case lower bounds under a prior over hypotheses.
>
> 9. The results may be only incremental over [2], since both use the Bayesian posterior and mutual information, with the main difference being that we use a 0–1-type loss while they use KL divergence. Are our bounds looser because of this choice (e.g., extra “$-1$” or square terms)? How would things change under other losses like KL or TV? Could Theorem 1 recover [2]’s bound if KL divergence were used?
>
> Our response is that although both works use a mutual-information term, [2] studies KL divergence between labeling functions while we study 0–1 classification error, making the two analyses mathematically incomparable and preventing Theorem 1 from recovering their result by simply changing the loss. [2]'s equalities rely on algebraic properties of KL divergence, whereas our bound fundamentally comes from Fano’s inequality, leading to different techniques, different quantities, and different conclusions. Moreover, our work provides additional insight—most notably the identifiability–agreement trade-off.
>
> [2] Hong Jun Jeon and Benjamin Van Roy. An information-theoretic framework for deep learning. In Alice H. Oh, Alekh Agarwal, Danielle Belgrave, and Kyunghyun Cho (eds.), Advances in Neural Information Processing Systems, 2022
>
> 10. Adding numerical studies can make the improvements in our examples more intuitive and to strengthen the paper.
>
> Our response is that we already conducted numerical analyses with closed-form solutions for all examples in our initial version—shown in Section 4.4 and Appendices A and B—which allowed us to compare bounds precisely and demonstrate the sharpness of our result.
>
> (Continued in the next comment.)

---

> > ### Author Response · Authors · 2025-12-02
> >
> > We would also like to hightlight several updates to the paper made in response to the reviewers’ suggestions.
> > * We have added a remark in the caption of figure 2.
> > * We have updated the reference of PAC lower bounds in line 36 which now cites [3].
> > * We have added a brief sketch of proof of Theorem 1 in Section 4.3.
> > * We have added a reference in line 67 to Remark 1 where we have explained the irrelevancy between KL divergence and 0-1 loss in classification problems.
> > *  We have added an explanation of the connection between Fano’s inequality and our lower bound and how it differs from prior work in Appendix E.
> > * We have corrected a typo in line 217: “unstrained” -> “unconstrained”.
> >
> > [3] Ehrenfeucht, A., Haussler, D., Kearns, M. and Valiant, L., 1989. A general lower bound on the number of examples needed for learning. Information and Computation, 82(3), pp.247-261.
> >
> > Finally, we once again thank the reviewers, the ACs, and the organizing committee for their time and effort throughout the review process.

---

### Meta-Review · Area_Chair_Yd6c · 2026-01-02

**Summary:**

This paper focuses on the hardness on learning in classification tasks. They consider a setting based on a fixed input distribution and a prior distribution on the set of classifier with an average notion of risk measure. They provide a closed-form solution for the optimal learner associated to the Bayes risk. They also provide information theoretic risk bounds that are tighter than other previous works.

Based on the initial reviews:

Reviewer 2Esk identifies as strengths:  clear and interpretable decomposition making a link with irreducible noise. The framing is consistent with Bayesian learning, and the result reads naturally. On the other hand: technical novelty limited (results follow from definitions or standard inequality), practical motivation insufficient.

For vCPZ, the paper is very accessible and clear, the decomposition in the lower bound bettween the “agreement” of hypotheses and their “identifiability”. The motivation in the first threshold / interval classification problem and the failure of prior work (Chen et al.) to capture this is very clear. On the other hand, examples did not provide cases when the bound can be loose, Thm1 would benefit from sketch proof for giving intuitions.

For vkM9, Strengths include a new clear formulation of standard classification problem which is interesting and worth exploring, the paper provides an "identifiability–agreement" tradeoff characterization of the hardness of learning. On the other hand: some reorganization is suggested and the results in Section are not intuitively understandable. Providing a practical interpretation of the framework would be useful.

s5Lw: Strengths: average case risk allows the work to avoid the pessimism of classical bounds with natural connection to practical distributions. Concrete examples of bound behavior. On the other hand: the presented results appears rather incremental wrt (Jeon & Roy, 22). Uncommon assumption for Thm2 that lacks proper justification, lack of numerical studies.

Overall, the paper presents a novel result with a tight risk bound. The setting has shown some interesting parts with a nice identifiability-agreement tradeoff. However, the most positive reviewers were clearly not experts, the others have some concerns about significance. If some reviewers may have increased their score, my point of view is that the novel parts of the paper have still to be refined and that the significance wrt other concurrent work must be strengthened. More expert reviewers were not that convinced with this version.
I propose then rejection.

**Reviewer Concerns:**

For Reviewer 2Esk: authors answer that their contribution lie on a novel problem formulation, an average risk analysis and identifiability-agreement trade-off, they show an example when the Bayes-risk is more informative than worst case perspective and discusses why their hypothesis is not that strong.

For vCPZ, authors did not give a case where the bound is loose and provided answers to other points.

For vkM9, authors have provided answers to all the points.

For s5Lw, authors have argued what makes their contribution different (different setting, different loss, new interpretation of involved quantities, identifiability-agreement tradeoff). The reviewer acknowledges these differences but finds them as relatively extension of the work of (Jeon&Roy)

**Reviewer Scores:**

Reviewer 2Esk gave a 4. Authors have provided direct answers to his concerns, not sure he would have updated his score, maybe a possible 5.

Reviewer vCPZ gave a 8 but identified himself as a non expert. He would probably have kept his score.

vkM9 gave a 4 but was not confident. It is unclear for me if he would have increased his score.

s5Lw gave a 2 and maintained his score.

---

### Decision · Program_Chairs · 2026-01-26

Reject